# What Do Latent Action Models Actually Learn?

**Chuheng Zhang**[*1]**, Tim Pearce**[*1]**, Pushi Zhang**[1]**, Kaixin Wang**[1]**, Xiaoyu Chen**[2]**,**
**Wei Shen**[3]**, Li Zhao**[1]**, Jiang Bian**[1]
[1]Microsoft Research [2]Tsinghua University [3]Independent Researcher
[*]Equal contribution: `zhangchuheng123@live.com`, `timpearce@microsoft.com`

## Abstract

Latent action models (LAMs) aim to learn action-relevant changes from unlabeled videos by compressing changes between frames as *latents*. However, differences between video frames can be caused by *controllable changes* as well as *exogenous noise*, leading to an important concern – do latents capture the changes caused by actions or irrelevant noise? This paper studies this issue analytically, presenting a linear model that encapsulates the essence of LAM learning, while being tractable. This provides several insights, including connections between LAM and principal component analysis (PCA), desiderata of the data-generating policy, and justification of strategies to encourage learning controllable changes using data augmentation, data cleaning, and auxiliary action-prediction. These findings are validated through numerical simulations, as well as experiments in more realistic settings. This investigation is the first to rigorously investigate how the structure of observations, actions, and noise influence LAM learning.

## 1 Introduction

Latent action models (LAMs) aim to infer controllable action changes from streams of image observations in an unsupervised manner (Rybkin et al., 2018; Menapace et al., 2021). This is valuable because action-labeled data is typically expensive to source, while unlabeled videos are abundant. Hence, it offers a route for embodied AI systems to learn from large unlabeled datasets, for instance using the inferred latent actions as targets for pre-training a policy, while a small amount of labeled data can be used to learn a mapping from latent to real action controls (Ye et al., 2024). This has proven effective in learning from videos of 2D video games, robotics, and even broadcast tennis footage (Menapace et al., 2021; Schmidt and Jiang, 2023; Bruce et al., 2024; Chen et al., 2024b; Ye et al., 2024; Sun et al., 2024; Cui et al., 2025; Gao et al., 2025).

The success of such LAM-based recipes relies on the inferred latent action labels mapping to the real control action signals of interest. However, there is concern that this may not always be the case. For example, there is an intuition that LAM's inferred latent 'actions' simply compress differences between consecutive frames, even when control actions are not the cause of those differences (McCarthy et al., 2024). As such, LAMs may only succeed in domains where the cause of changes between observations can be fully attributed to the control action. A further point of dispute is whether a bottleneck is required (Schmidt and Jiang, 2023).

To study these issues and more, this paper conducts a theoretical analysis of a linear version of LAM. By retaining the architecture of recent LAM models, but swapping deep neural network components with simpler linear layers, we preserve the fundamental challenge of LAM training in an analytically tractable form. Our analysis of linear LAM firstly provides precise insights into when inferred latent actions capture true control signals compared to noise, as well as what information is captured by different components within LAM. Surprisingly, our analysis also reveals additional issues not currently known to the LAM community – related to the over-parametrization property of LAM and the randomness of data-generation policy. Finally, we propose and study potential solutions to ameliorate these issues – data augmentation and predicting action as an auxiliary task.

39th Conference on Neural Information Processing Systems (NeurIPS 2025).

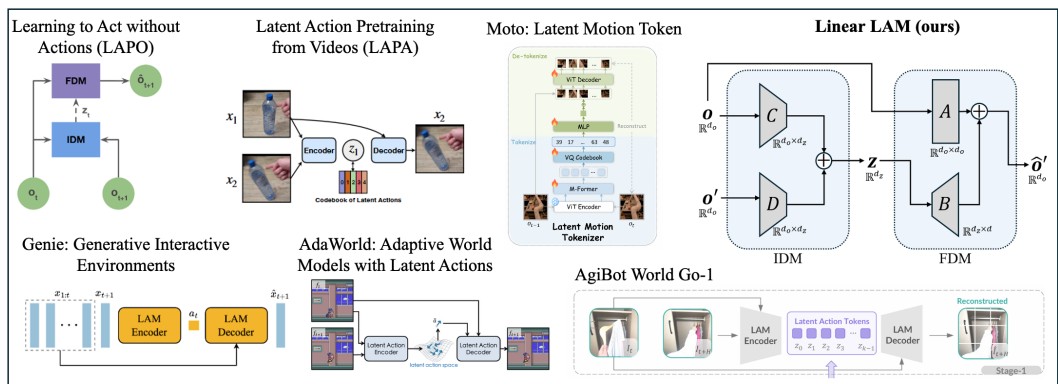

Figure 1: Linear LAM is an abstraction of the LAMs used in previous work. Inputting consecutive observation pairs $(o, o')$, the LAMs output the second observation via a reconstruction loss, $\|\hat{o}' - o'\|_2^2$. An information bottleneck tries to stop the direct copying of $o'$, with the expectation the latent $z$ will correspond to the control action $\mathbf{a}$. Linear LAM captures the essence of LAM training whilst being analytically tractable. The diagrams of previous LAMs are copied from their original papers: LAPO (Schmidt and Jiang, 2023), LAPA (Ye et al., 2024), Moto (Chen et al., 2024c), Genie (Bruce et al., 2024), AdaWorld (Gao et al., 2025), and Go-1 (AgiBot-World, 2025).

Concretely, this paper makes several key contributions.

1. Section 3 presents linear LAM, a tractable model preserving the essence of LAMs used in practice.

2. Section 4.1 shows linear LAM reduces to principal component analysis (PCA) on a mixture of controllable changes and exogenous noise, under certain assumptions. Our analysis justifies the practical use of LAM when the controllable action signals cause larger changes to observations than the exogenous noise.

3. Section 4.2 shows correlation between observations and actions decreases LAM's focus on learning controllable changes. This suggests that higher randomness in data-generating policies benefits LAM's learning.

4. Section 4.3 validates that performing data augmentation during LAM training can mitigate the over-parametrization issue and thus improve the semantics of the latent.

5. Section 4.4 finds that adding an action-prediction head encourages LAM to prioritize the learning of controllable changes for the latent.

6. Section 5 verifies that the main findings based on linear LAM still hold on more realistic LAMs.

## 2 Related Work

The study of learning representations of real actions in reinforcement learning (RL) has a long history. For instance, PG-RA (Chandak et al., 2019) clusters actions based on the similarity of their impact on the state to improve generalization in the action space. LASER (Allshire et al., 2021) learns latent actions through an encoder-decoder architecture trained to reconstruct real actions, resulting in higher learning efficiency in RL. TAP (Jiang et al., 2022) learns the latent action that can help to reconstruct the full trajectory (with state, action, and reward) condition on the state. EAR (Hua et al., 2022) finds that the latent task embedding resulting from the training of multi-task policies turn out to be good action representations with a geometrically and semantically meaningful structure. AD3 (Wang et al., 2024) adopts an inverse dynamics model and a forward dynamics model to extract latent action, similar to popular LAMs, but conditions these models on the real actions.

While the above papers learn action representations based on real actions, our paper focuses on learning latent actions *without access to the real action labels*. Removing the need for action labels during training is advantageous as it allows leveraging internet-scale video datasets in the pre-training stage (Miech et al., 2019; Chen et al., 2024a; Pei et al., 2025; Wang et al., 2023) – for example unlabeled demonstrations of humans completing diverse tasks. While high-quality robotic datasets with action annotations exist (Vuong et al., 2023; Fang et al., 2023; Khazatsky et al., 2024; AgiBot-World, 2025), they remain limited in scale. Such datasets can be integrated into the

semi-supervised learning framework to extract latent actions (Nikulin et al., 2025) or the policy fine-tuning stage (Schmidt and Jiang, 2023; Ye et al., 2024). An alternative approach aims to extract pre-defined actions from observations using computer vision techniques (Mendonca et al., 2023).

Beginning from Rybkin et al. (2018), LAMs have featured an information bottleneck or auto-encoder to allow learning in an unsupervised manner. ILPO (Edwards et al., 2019) learns latent actions along with the training of a policy that outputs the latent action and a world model that conditions on the latent action. Menapace et al. (2021) proposes a probabilistic action network that extracts a discrete action label and a continuous action variability embedding from consecutive observations. This network is trained jointly with an action decoder to generate video controlled by extracted actions. While these works adopt a bottleneck in the learning of latent actions, their training losses are complicated by involving policy learning or recurrent networks. LAPO (Schmidt and Jiang, 2023) proposes a LAM design with an inverse dynamics model extracting a latent action and a forward dynamics model that reconstructs the next observation based on the latent action. This forms the template of the modern LAM. Many papers follow a similar architectures to LAPO with a discrete latent action space such as FICC (Ye et al., 2022), Genie (Bruce et al., 2024), LAPA (Ye et al., 2024), Moto (Chen et al., 2024c), IGOR (Chen et al., 2024b), Go-1 (AgiBot-World, 2025), and GR001T N1 (Nvidia, 2025). However, there is a debate whether discrete latent actions are better than continuous latent actions. For example, Nikulin et al. (2025) finds that using continuous latent actions with a larger bottleneck can not only result in better predictability on real actions but also lead to better performance for downstream policies. We consider continuous latent actions in our linear LAM analysis while vector quantized latents are tested empirically in Section 5.

In contrast to the popularity of LAMs in recent foundation models for embodied AI, issues around the objective, learnability, and robustness of LAM have received limited attention. Consequently, our paper aims to investigate these issues.

## 3 Setup

This section broadly introduces the problem setting, goal and model used in recent LAM work in practice. We then more formally detail these for the linear model to be used in subsequent analysis. Finally, we outline the details of our simulation setup to be used in support of our later analysis.

### 3.1 LAMs in Practice

**Setting.** The setting tackled by recent LAM work assumes access to a large dataset of pairs of observations and next observations $\mathcal{D} = \{(o_i, o_i')\}_{i=1}^{N}$ and only a small subset of it has action labels $\mathcal{D}_a = \{(o_i, o_i', a_i)\}_{i=1}^{N_a}$, with $\lambda := |\mathcal{D}_a|/|\mathcal{D}| \ll 1$. Note we drop the $i$ subscript when we do not need to refer to specific samples. In practice, $o$ and $o'$ could be feature vectors extracted from the original observations (Cui et al., 2025), and $o$ could be a stack of historical observations to account for partial observability (Bruce et al., 2024). The two datasets are assumed to come from the same distribution in most LAM work (Menapace et al., 2021; Bruce et al., 2024; Ye et al., 2024)[1].

**Model.** Recent LAM designs (Schmidt and Jiang, 2023; Bruce et al., 2024; Ye et al., 2024; Chen et al., 2024b; AgiBot-World, 2025) (see Figure 1) take a pair of consecutive observations $(o, o')$ as input, and output a latent $z$ as well as a prediction of the next observation $\hat{o}'$. (We subsequently avoid calling $z$ 'latent action' to avoid confusion with the real action.)

LAMs are typically decomposed into an inverse dynamics model (IDM) and forward dynamics model (FDM), implemented as deep neural networks, and trained via a reconstruction loss,

$$z = \psi_{\text{IDM}}(o, o'), \quad \hat{o}' = \psi_{\text{FDM}}(o, z), \quad \mathcal{L} := \min_{\psi_{\text{IDM}}, \psi_{\text{FDM}}} \mathbb{E}_{\mathcal{D}}\big[\|\hat{o}' - o'\|_2^2\big]. \tag{1}$$

where $\psi_{\text{IDM}}$ contains a bottleneck so the dimension of $z$ is smaller than that of $o$.

**Use cases.** We identify two primary downstream use cases of LAMs.

- The LAM latents can be used as *input* to a world model $\hat{T}$ trained to generate future frames, $\hat{T}(o'|o, z)$ (Sun et al., 2024; Chen et al., 2024b; Bruce et al., 2024; Menapace et al., 2021). These

---

[1]Certain work explores mismatch case, e.g., $\mathcal{D}$ is human videos and $\mathcal{D}_a$ is robotics data (Chen et al., 2024b).

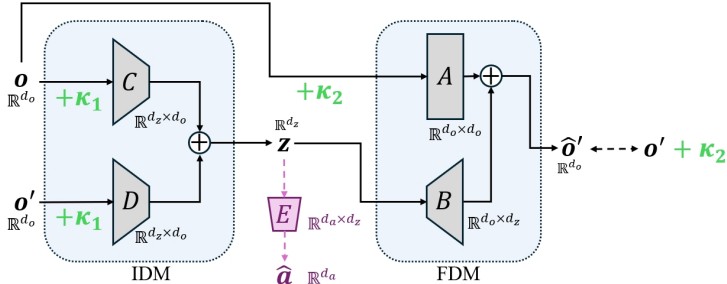

Figure 2: Overview of linear LAM. Grey blocks represent learnable parameter matrices, giving rise to the predictive model $\hat{\boldsymbol{o}}' = A\boldsymbol{o} + B(C\boldsymbol{o} + D\boldsymbol{o}')$. Green illustrates linear LAM with data augmentation to reduce the amount of information the latent contains about the observation (in Section 4.3). Pink illustrate linear LAM with auxiliary action prediction to encourage the latent to focus on the controllable actions and suppress the noise signal (in Section 4.4).

world models can generate higher-quality frames than the FDM in LAM, but have been used to only qualitatively interpret the meaning of the learned latents.

- The LAM latents can be used as *labels* in the pre-training of a latent policy $\pi_{\text{latent}}(\hat{\boldsymbol{z}}|\boldsymbol{o})$ (similar to behavior cloning on actions). This policy may later be mapped to real actions, $\pi_{\text{map}}(\hat{\boldsymbol{a}}|\hat{\boldsymbol{z}})$ (Bruce et al., 2024; Schmidt and Jiang, 2023), or be followed by a fine-tuning phase on real actions (Schmidt and Jiang, 2023; Ye et al., 2024; Chen et al., 2024b).

In both use cases, it is hoped that the learned latents can be aligned with the true actions as closely as possible. As in Schmidt and Jiang (2023), "*our hypothesis is ... [this] may allow us to learn a latent action representation with a structure closely corresponding to the true action space*".

## 3.2 Linear LAM

**Setting.** We conduct analysis in the controlled Markov process (CMP) framework (Puterman, 2014) with $(\mathcal{O}, \mathcal{A}, T)$. At a given timestep, the agent receives observation $\boldsymbol{o} \in \mathcal{O}$ and takes action $\boldsymbol{a} \in \mathcal{A}$. The transition function describes the probability distribution over the next observation, $T(\boldsymbol{o}'|\boldsymbol{o}, \boldsymbol{a})$.

We consider vector observations, $\mathcal{O} = \mathbb{R}^{d_o}$ (which could be thought of as image observations processed with a pre-trained image encoder as in Cui et al. (2025); Chen et al. (2024b)). The action space $\mathcal{A} = \mathbb{R}^{d_a}$ has lower dimensionality than $\mathcal{O}$, i.e., $d_a \ll d_o$[2]. These actions are mapped to create *controllable changes* in observation space via an *action effect matrix* $X \in \mathbb{R}^{d_o \times d_a}$, $\boldsymbol{q} = X\boldsymbol{a}$.

We generally assume linear LAM cannot access the action $\boldsymbol{a}$ during the training, but we will make use of it to evaluate the learned LAM in simulations. We choose the transition function to be an additive combination of *controllable changes* $\boldsymbol{q} \in \mathbb{R}^d$ (the state changes caused by the ego agent's actions) and *exogenous noise* $\boldsymbol{\epsilon} \in \mathbb{R}^d$ (representing environmental stochasticity or other agent's actions), i.e. with the next state generated via, $\boldsymbol{o}' = \boldsymbol{o} + \boldsymbol{q} + \boldsymbol{\epsilon}$.

**Model.** For linear LAM, the IDM and FDM consist of linear mappings. Summarized in Figure 2, the IDM and FDM are given by,

$$\boldsymbol{z} = \psi_{\text{IDM}}^{\text{Linear}}(\boldsymbol{o}, \boldsymbol{o}') \coloneqq C\boldsymbol{o} + D\boldsymbol{o}' \tag{2}$$

$$\hat{\boldsymbol{o}}' = \psi_{\text{FDM}}^{\text{Linear}}(\boldsymbol{o}, \boldsymbol{z}) \coloneqq A\boldsymbol{o} + B\boldsymbol{z} = A\boldsymbol{o} + B(C\boldsymbol{o} + D\boldsymbol{o}'), \tag{3}$$

where $\boldsymbol{z} \in \mathcal{Z} = \mathbb{R}^{d_z}$ with $d_z \ll d_o$ is the latent. All matrices are learnable parameters, including FDM parameters $A \in \mathbb{R}^{d_o \times d_o}$, $B \in \mathbb{R}^{d_o \times d_z}$, and IDM parameters $C, D \in \mathbb{R}^{d_z \times d_o}$.

As for practical LAM (Eq. 1), the linear LAM is trained via a reconstruction loss,

$$\mathcal{L}(A, B, C, D) = \mathbb{E}\left[\|\hat{\boldsymbol{o}}_i' - \boldsymbol{o}_i'\|_2^2\right]. \tag{4}$$

Appendix A summarizes the gap between linear and practical LAM.

---

[2]Discrete actions can be represented as one-hot vectors within $\mathbb{R}^{d_a}$. While most of analysis (except that on bottleneck) should also apply to discrete actions, we confine our analysis in continuous actions.

**Goal.** Following our discussion of use-cases of LAM cases in Section 3.1, we interpret the desired latents to contain as much information as possible about the real action, while minimizing the amount of information about the exogenous noise and the observation. This can be formalized as,

$$\text{LAM Objective} := \max_{\boldsymbol{z}} \left[ \mathcal{I}(\boldsymbol{z}; \boldsymbol{a}) - \mathcal{I}(\boldsymbol{z}; \boldsymbol{\epsilon}) - \mathcal{I}(\boldsymbol{z}; \boldsymbol{o}) \right], \tag{5}$$

where $\mathcal{I}$ is mutual information. Hence, an ideal latent should contain all information about the action label, and no other parts of the environment, consistent with intuitions of previous work – *"biasing [the latent] towards simpler representations is likely preferrable"* (Schmidt and Jiang, 2023).

Whilst mutual information is typically a challenging quantity to measure, in our linear model we can approximate this straightforwardly. After training the linear LAM via Eq. 4 and freezing the parameters, we fit three additional linear layers predicting $(\boldsymbol{q}, \boldsymbol{\epsilon}, \boldsymbol{o})$ from $\boldsymbol{z}$, to give $(\hat{\boldsymbol{q}}, \hat{\boldsymbol{\epsilon}}, \hat{\boldsymbol{o}})$ respectively. Note that, under the assumption that the variables $\mathbf{x}$ and $\mathbf{y}$ are independent multivariate Gaussian variables, the mutual information $\mathcal{I}(\mathbf{x}, \mathbf{y}) = -\frac{1}{2} \log \left( \frac{\|\hat{\mathbf{y}} - \mathbf{y}\|_2^2}{\mathbb{V}\text{ar}(\mathbf{y})} \right)$ where $\hat{\mathbf{y}}$ is the least squares estimate (LSE) of $\mathbf{y}$ based on $\mathbf{x}$, and $\mathbb{V}\text{ar}(\cdot)$ indicates the total variance (see e.g., Chapter 8 of (Johnson et al., 2002)) of a multivariate random variable. Hence, we can define an evaluation metric that captures the objective of training linear LAM,

$$\text{Linear LAM Objective (LLO)} := \max_{\boldsymbol{z}} \left[ -\frac{\|\hat{\boldsymbol{q}} - \boldsymbol{q}\|_2^2}{\mathbb{V}\text{ar}(\boldsymbol{q})} + \frac{\|\hat{\boldsymbol{\epsilon}} - \boldsymbol{\epsilon}\|_2^2}{\mathbb{V}\text{ar}(\boldsymbol{\epsilon})} + \frac{\|\hat{\boldsymbol{o}} - \boldsymbol{o}\|_2^2}{\mathbb{V}\text{ar}(\boldsymbol{o})} \right]. \tag{6}$$

Note that we get rid of the $\log(\cdot)$ function wrapped around the MSE loss to better calculate the values for this objective since the value range for the mutual information $[0, +\infty)$ is unbounded. This objective is maximized when $\boldsymbol{z}$ perfectly predicts $\boldsymbol{q}$ while containing no information about $\boldsymbol{\epsilon}$ and $\boldsymbol{o}$, resulting in the optimal value for LLO equal to $0 + 1 + 1 = 2$.

# 4 Analysis

At times, we will present numerical simulations of linear LAM to visually communicate later analysis. See the details of simulation in Appendix B and the code in supplementary.

## 4.1 Analysis 1: Linear LAM is PCA

This section first shows that training linear LAM is equivalent to performing PCA on the mixture of controllable changes and exogenous noise. This requires that the controllable changes and exogenous noise $\boldsymbol{q}, \boldsymbol{\epsilon}$ are uncorrelated with the observation $\boldsymbol{o}$ (Section 4.2 relaxes this assumption).

We discuss the insight that this connection provides, followed by an analysis of several important cases covering specific settings of controllable changes and exogenous noise.

**Proposition 4.1** (Linear LAM is PCA). *Under the linear LAM model and setup defined in Section 3.2, and additionally assuming $\mathbb{E}[\boldsymbol{o}(\boldsymbol{q} + \boldsymbol{\epsilon})^T] = \boldsymbol{0}$, the objective of linear LAM is equivalent to performing PCA on a mixture of controllable changes $\boldsymbol{q}$ and exogenous noise $\boldsymbol{\epsilon}$,*

$$\mathcal{L} = \mathbb{E} \left[ \|(BD - I)(\boldsymbol{q} + \boldsymbol{\epsilon})\|_2^2 \right] \tag{7}$$

Note, $BD$ is a low-rank matrix to capture the main components of $\boldsymbol{q} + \boldsymbol{\epsilon}$. See proof in Appendix C.1.

Given that we have transformed the optimization problem of linear LAM into a PCA problem (which can also be viewed as a linear auto-encoder), PCA's property applies to linear LAM.

**Proposition 4.2** (Linear LAM tries to capture $\boldsymbol{q} + \boldsymbol{\epsilon}$). *Denote the covariance matrix of $\boldsymbol{q} + \boldsymbol{\epsilon}$ as $\Sigma_{\boldsymbol{q}+\boldsymbol{\epsilon}} = \mathbb{E}[(\boldsymbol{q} + \boldsymbol{\epsilon})(\boldsymbol{q} + \boldsymbol{\epsilon})^T]$ and its eigenvalue decomposition $\Sigma_{\boldsymbol{q}+\boldsymbol{\epsilon}} = U \Lambda U^T$ with eigenvalues $\lambda_1 \geq \lambda_2 \geq \cdots \geq \lambda_{d_o}$. Under the same conditions of Proposition 4.1, $\mathcal{L}$ is optimized when*

1. *$B = U_{d_z}$ spans the subspace of the top $d_z$ principal components of $\Sigma_{\boldsymbol{q}+\boldsymbol{\epsilon}}$ where $U_{d_z}$ contains the first $d_z$ columns of $U$;*

2. *$D = U_{d_z}^T$ such that the reconstruction $BD(\boldsymbol{q} + \boldsymbol{\epsilon}) = U_{d_z} U_{d_z}^T (\boldsymbol{q} + \boldsymbol{\epsilon})$ projects $\boldsymbol{q} + \boldsymbol{\epsilon}$ onto the subspace spanned by $U_{d_z}$.*

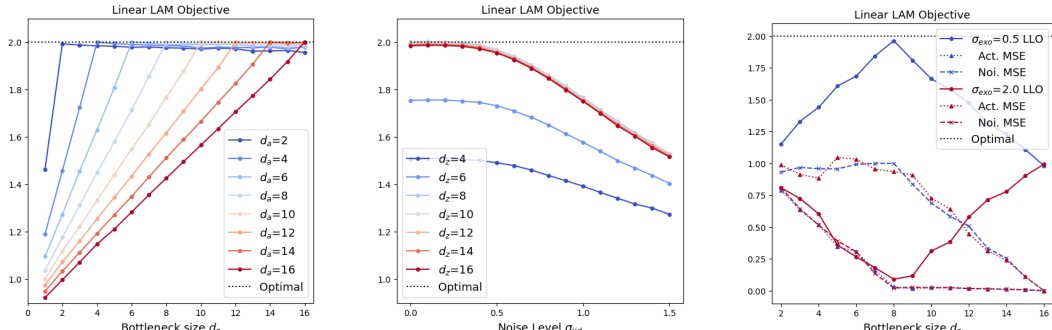

Figure 3: LLO (Linear LAM objective (6), higher better) measured in three noise settings. (Left) $\epsilon = 0$. (Middle) $\epsilon$ is i.i.d. noise. (Right) $\epsilon$ contains the effect of other agents. Action MSE, noise MSE, and observation MSE are the three terms in (6). We set real action dimension $d_a = 8$ and exogenous action dimension $d_b = 8$ unless otherwise stated, and ensure $q$ has unit variance.

*The minimum loss is given by the sum of the eigenvalues of corresponding to the discarded principal components $\mathcal{L}^* = \sum_{i=d_z+1}^{d_o} \lambda_i$.*

This proposition is equivalent to the Eckart–Young–Mirsky theorem (Eckart and Young, 1936) whose proof can be found in Chapter 2.4 of Golub and Van Loan (2013).

**Over-parameterization issue.** We show that linear LAM is over-parameterized, with multiple solutions of $(A, B, C, D)$ able to minimize the reconstruction loss objective. Specifically, the latent $z = Co + Do'$, in addition to capturing information about $q$ and $\epsilon$, may further contain information about $o$. There is no detrimental effect provided the $A$ matrix compensates to 'knock out' $o$'s information in $z$. Concretely, there exists a family of solutions $B(Co + Do') = (q + \epsilon) + \alpha o$ and $A = (1 - \alpha)I$ for any $\alpha \in \mathbb{R}$ such that $\hat{o}' = B(Co + Do') + Ao = (q + \epsilon) + (1 - \alpha)o + \alpha o = o'$.

We will revisit this issue in Section 4.3, showing that data augmentation handles this over-parameterization issue. For the purpose of our immediate analysis, we predict $(\hat{q}, \hat{\epsilon}, \hat{o})$ from a surrogate latent $\tilde{z} := B^{-1}(\hat{o}' - o)$ when calculating LLO to get around this issue. This surrogate latent is the same as the original one when $A = I$, which is the case when data augmentation is adopted. Hence, we have $C = -D$ which indicates that the semantic meaning of the latent $z = Co + Do'$ remains the same (*no movement*) when $o' = o$ across different observations.

**Case 1: $\epsilon = 0$.** In the absence of exogenous noise $\epsilon$, linear LAM does capture the true action $a$ in the latent $z$, when the bottleneck dimension is set equal or larger than the action dimension.

When $\epsilon = 0$ and $q = Xa$, the covariance matrix of $\Sigma_{q+\epsilon} = \Sigma_q = \mathbb{E}[Xaa^TX]$ only has $d_a$ non-zero eigenvalues. Following Proposition 4.2, we can make the following conclusions.

- When $d_z \geq d_a$, the capacity of the latent is large enough to capture all the information about the controllable change $q$, the minimum loss $\mathcal{L}^* = 0$, and LLO achieves the optimal.

- Specifically, when $d_z = d_a$, the subspace spanned by the columns of $B = U_{d_a}$ is the same to the subspace spanned by the columns of action effect matrix $X$ (by noting that $U_{d_a}\Lambda U_{d_a}^T = \Sigma_{q+\epsilon} = X\mathbb{E}[aa^T]X^T$). In this case, the learned latent $z$ (whose effect is interpreted by $B$) fully captures the information of $a$ (whose effect is $X$), and LLO is maximized. In other words, for this ideal case linear LAM's latent perfectly captures the information of the true action $a$ without access to it.

- When $d_z < d_a$, $\mathcal{L}^* > 0$ and this linear auto-encoder captures the first $d_z$ components in $q$.

We illustrate how LLO varies across different $d_a$ and $d_z$ through numerical simulation in Figure 3 (left). The simulation validates that linear LAM is optimized in terms of LLO when $d_z \geq d_a$.

**Case 2: $\epsilon$ is i.i.d. noise.** We consider i.i.d. noise (independent and identically distributed), which may be a realistic assumption in the case of sensors or image encoders. We assume $\epsilon$ is i.i.d. with zero-mean $\mathbb{E}[\epsilon] = 0$ and isotropic covariance $\mathbb{E}[\epsilon\epsilon^T] = \sigma_{iid}^2 I$. Considering that $q$ and $\epsilon$ are independent, the covariance matrix $\Sigma_{q+\epsilon}$ can be eigenvalue decomposed as $\Sigma_{q+\epsilon} = U\Gamma_0 U^T = U_0(\Lambda_0 + \sigma_{iid}^2 I)U_0^T$ where $U_0$ and $\Lambda_0$ are the eigenvectors and eigenvalues of $\Sigma_q$ respectively (i.e., when $\epsilon = 0$). Combining the results in Proposition 4.2, we conclude that, when there is i.i.d. noise, 1) the FDM

parameter $B$ (and therefore the semantics of the latent since $B$ interpret the latent) remains the same, and 2) the loss increases since the eigenvalues increase. This conclusion is consistent with the conclusion on the robustness of PCA (Anderson, 1963; Johnstone, 2001).

Figure 3 (middle) shows how LLO changes for Gaussian noise of differing variance. We observe that linear LAM is robust to i.i.d. noise up to around $\sigma_{iid} = 0.5$ (when the noise intensity is half that of the signal), linear LAM still succeeds with negligible gap to the optimal case.

**Case 3: $\epsilon$ contains the effect of other agents.** In many real-world datasets (such as Ego4d (Grauman et al., 2022)), the change between two observations may not only be caused by the control action of the ego-agent (e.g. joints of a robot arm), but additionally can be effected by exogenous noise, such as 'other agents' (e.g. a person walking in the background, camera shake). Compared with i.i.d. noise, this noise is structured and would be predictable if the other agent's control actions were available.

Here, we assume $\epsilon = Y\mathbf{b}$ where $Y \in \mathbb{R}^{d_o \times d_b}$ is the exogenous effect matrix and $\mathbf{b}$ represents the action taken by other agents. In this case, analysis similar to that of Case 1 concludes that *linear LAM will learn to capture effects with largest variances, no matter if they result from the controllable action or other agents.* Moreover, when the columns in $Y$ are not orthogonal to that of $X$, the FDM parameter $B$ will be impacted by the exogenous noise.

Simulation results in Figure 3 (right) illustrate which part of information (the controllable $q$ or the noise $\epsilon$) enters the latent when the latent dimension $d_z$ is increased. We observe that: 1) When the variance of exogenous noise is smaller than $q$ (i.e., $\sigma_q = 1 > \sigma_{exo} = 0.5$), linear LAM learns to fit the controllable part first, and $d_z = d_a$ is still the optimal configuration. 2) When the variance of noise exceeds that the signal $q$ (i.e., $\sigma_q = 1 < \sigma_{exo} = 2.0$), linear LAM will first fit the noise, and the latent fails to exclude the information of the noise no matter how we set $d_z$. To alleviate this issue, an obvious solution in practice is to preprocess training data to reduce significant noise components (such as stabilizing camera shake).

## 4.2 Analysis 2: Effects of Data Collection Policy

Section 4.1 considered the case where both the controllable changes and the exogenous noise are independent of the observation, i.e., $\mathbb{E}[\mathbf{o}(\mathbf{q} + \boldsymbol{\epsilon})^T] = \mathbf{0}$. However, practical LAMs are usually trained on data created by expert policies, e.g., robotic datasets often consist of human teleoperation. Within such datasets, the observation *is* correlated with the action $\mathbf{a}$, and thus the controllable change $\mathbf{q}$, resulting in $\mathbb{E}[\mathbf{o}(\mathbf{q} + \boldsymbol{\epsilon})^T] \neq \mathbf{0}$. This section analyzes this case in linear LAM.

Specifically, we assume the data generating policy acts via $\mathbf{a} = \Pi_d \mathbf{o} + \boldsymbol{\pi}_s$ where $\Pi_d \in \mathbb{R}^{d_a \times d_o}$ represents the deterministic part of the policy and $\pi_s \in \mathbb{R}^{d_a}$ represents the stochastic part of the policy. We assume that $\mathbf{o}$, $\boldsymbol{\epsilon}$, and $\boldsymbol{\pi}_s$ are uncorrelated, but there will be correlation between $\mathbf{o}$ and $\mathbf{a}$.

In this case, since $\mathbb{R}[\mathbf{oq}^T] \neq 0$, the cross term in the expansion of this loss cannot be ignored, and no longer reduces to PCA. Letting $\Sigma_{\mathbf{o}} := \mathbb{E}[\mathbf{oo}^T]$ be the covariance matrix of the observation, we can further solve for $A$ by setting the partial derivative of the loss function w.r.t. $A$ to zero, obtaining

$$A = I - (BC + BD) - \Sigma_{\mathbf{o}}\Pi_d^T X^T (BD - I)^T \Sigma_{\mathbf{o}}^{-1}. \tag{8}$$

This differs from the previous result for the random policy where $A = I - (BC + BD)$ by the last term $\Sigma_{\mathbf{o}}\Pi_d^T X^T (BD - I)^T \Sigma_{\mathbf{o}}^{-1}$. The intuitive interpretation for this term is that *A will capture changes caused by the deterministic part of the policy* (noting that $BD - I$ can project the vectors onto the orthogonal complement of the column space of $B$). This is problematic for LAM, since the FDM can implicitly absorb deterministic parts of the policy, leaving capacity in the latent available to absorb exogenous noise. Intuitively, if action $\mathbf{a}^*$ is *always* selected for observation $\mathbf{o}^*$, this *always* leads to $\mathbf{o}'^*$. The FDM can simply learn that the mapping $\mathbf{o}^* \rightarrow \mathbf{o}'^*$, without the need for a latent $\mathbf{z}$.

In our simulation, we control the randomness of the policy using a coefficient $\chi$, and let $\mathbf{a} = \chi\Pi_d\mathbf{o} + (1 - \chi)\boldsymbol{\pi}_s$ where $\Pi_d$ is a random projection matrix and $\boldsymbol{\pi_s}$ is a standard Gaussian vector. The simulation results in Figure 4 (left) indicate that the more deterministic the data collection policy is, the less information about $\mathbf{a}$ is captured by $\mathbf{z}$. Therefore, *the analysis in this section suggests data collection policies with higher randomness are preferred for LAM training.* Trajectories collected by deterministic expert policies may lead to poor quality LAMs.

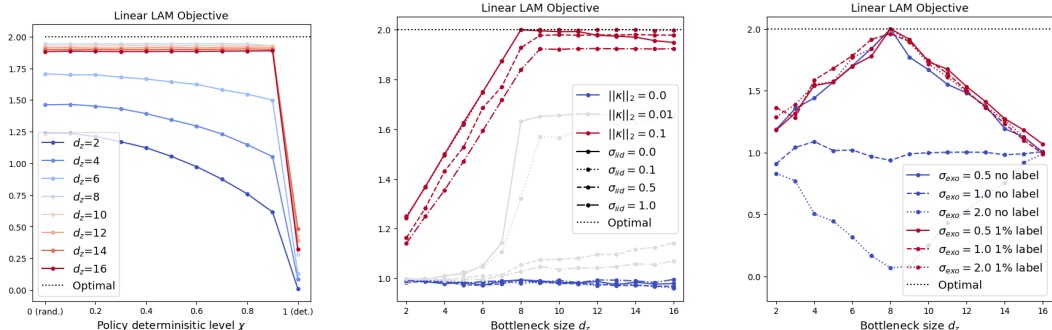

Figure 4: LLO (6) in three settings. (Left) Data generated from policies ranging from fully random to fully deterministic. (Middle) Linear LAM trained w/ varying strength data augmentation. (Right) Linear LAM trained w/ and w/o action prediction. We set $d_a = 8$ in the experiments.

### 4.3 Analysis 3: Improvements via Data Augmentation

This section analyzes the augmentation scheme used in Chen et al. (2024b), which will turn out to resolve a key issue encountered in linear LAM (see the discussion on over-parameterization in Section 4.1), where information about the observation $\boldsymbol{o}$ enters into the latent $\boldsymbol{z}$.

The IGOR model (Chen et al., 2024b) applies one shared random crop to both inputs of the IDM, and a second shared random crop applied to the FDM and reconstruction target. The intuition is that "*by using different croppings [for IDM and FDM], the model is encouraged to learn a more semantically invariant latent action*". Our subsequent analysis shows this intuition is provably correct in linear LAM, resulting in new terms to the loss that encourage removal of the latent of any semantic information about the observation. Sun et al. (2024) also apply a similar scheme, with a single action preserving crop applied to the IDM which makes it harder "*to copy the appearance information directly from the [IDM]*".

**Data augmentation in linear LAM.** We extend our linear LAM setup to include a data augmentation operator $\mathrm{Aug}_i[\boldsymbol{o}] \coloneqq \boldsymbol{o} + \boldsymbol{\kappa}_i$ that takes an observation as input and adds some random vector $\boldsymbol{\kappa} \in \mathbb{R}_o^d$. $\mathrm{Aug}_i[\cdot]$ applies the same $i$-th operator to different variables, say $\mathrm{Aug}_1[\boldsymbol{o}]$ and $\mathrm{Aug}_1[\boldsymbol{o}']$ will apply a consistent random variable $\boldsymbol{\kappa}_1$ to each observation.

**Proposition 4.3** (Data augmentation addresses over-parameterization)**.** *With data augmentation,*

$$\boldsymbol{z} = \psi_{IDM}^{Linear}(\mathrm{Aug}_1[\boldsymbol{o}], \mathrm{Aug}_1[\boldsymbol{o}']_1) \coloneqq C(\boldsymbol{o} + \boldsymbol{\kappa}_1) + D(\boldsymbol{o}' + \boldsymbol{\kappa}_1) \tag{9}$$

$$\hat{\boldsymbol{o}}' = \psi_{FDM}^{Linear}(\mathrm{Aug}_2[\boldsymbol{o}], \boldsymbol{z}) \coloneqq A(\boldsymbol{o} + \boldsymbol{\kappa}_2) + B\boldsymbol{z}. \tag{10}$$

*and assuming $\mathbb{E}[\boldsymbol{o}(\boldsymbol{q} + \boldsymbol{\epsilon})^T] = \boldsymbol{0}$, optimizing the loss defined in* (1) *results in $A = I$ and $C + D = 0$.*

See Appendix C.2 for proof. What is the effect of encouraging $A = I$ and $C + D = 0$? If, $A$ is the identity, all observation information flows directly through this matrix, and $\boldsymbol{z}$ need not carry any additional information about $\boldsymbol{o}$ (given our additive transition function). Equivalently, setting $C + D = 0$ cancels out $\boldsymbol{o}$ information in $\boldsymbol{z}$.

$$\boldsymbol{z} = C(\boldsymbol{o} + \boldsymbol{\kappa}_1) + D(\boldsymbol{o}' + \boldsymbol{\kappa}_1) = -D(\boldsymbol{o} + \boldsymbol{\kappa}_1) + D(\boldsymbol{o} + \boldsymbol{q} + \boldsymbol{\epsilon} + \boldsymbol{\kappa}_1) = D(\boldsymbol{q} + \boldsymbol{\epsilon}) \tag{11}$$

This condition improves the semantic meaning of the latent action across different observations. For example, the zero latent $\mathbf{z} = \boldsymbol{0}$ has the semantic meaning *"the frame does not change from $\boldsymbol{o}$ to $\boldsymbol{o}'$"*, which is consistent across different observations. Hence, this augmentation scheme is a mechanism to remove information about $\boldsymbol{o}$ from $\boldsymbol{z}$, explicitly minimizing $\mathcal{I}(\boldsymbol{z}; \boldsymbol{o})$ in the original objective (5).

**Simulation.** We show the effect of data augmentation under different noise levels in Figure 4 (middle). Starting from here, the simulations calculate LLO based on the true latent $\boldsymbol{z}$ instead of the pseudo-latent. Due to this change, even when there is no noise $\sigma_{iid} = 0$, the learned latent $\boldsymbol{z}$ cannot perfectly predict the real action $\boldsymbol{a}$ without other information. However, adding an augmentation vector with 0.1 magnitude variance (10% of the observation variance), greatly improves the latent learned by linear LAM, achieving close to the ideal LLO.

**Designing data augmentation.** In our linear setting, we have designed the data augmentation $\boldsymbol{\kappa}$ as additive and i.i.d. across its elements. This design is based on our knowledge that the semantic

meaning for the frame change should be *invariant* to this data augmentation, i.e., $\boldsymbol{o}' - \boldsymbol{o} = (\boldsymbol{o}' + \boldsymbol{\kappa}) - (\boldsymbol{o} + \boldsymbol{\kappa})$, since the dynamics are also additive. For real images, Chen et al. (2024b) adopt random crops, reflecting a prior that the action is *invariant* to position (e.g., a robot closing its gripper should produce the same latent action regardless of its location in the image). Further, the variance of different augmentations determines how important this term is – a larger augmentation variance enforces this constraint more strictly. In summary, our analysis justifies the practice of designing data augmentations that capture a model designer's domain knowledge about desired invariances to improve the semantics of the learned latent.

### 4.4 Analysis 4: Improvements via Auxiliary Action Prediction

This section analyzes the setting when a small dataset of action labeled data $\mathcal{D}_a$ is available during training of the LAM. This can be used to help guide the latents to represent controllable changes rather than exogenous noise. Specifically we consider the a simple auxiliary loss, with latents as input, predicting the action labels when available. Nikulin et al. (2025) also provide empirically evidence to show that this is a promising strategy to avoiding focus on 'distractors' present in the observations.

**Action prediction in linear LAM.** Consider a linear prediction head $E \in \mathbb{R}^{d_a \times d_z}$ at the latent bottleneck, $\hat{\boldsymbol{a}} = E\boldsymbol{z}$, and a corresponding action reconstruction loss $\|\hat{\boldsymbol{a}} - \boldsymbol{a}\|_2^2$, we optimize this objective, $\mathcal{L}_a := \mathbb{E}_{\mathcal{D}} \left[ \|\hat{\boldsymbol{o}}' - \boldsymbol{o}'\|_2^2 \right] + \lambda \mathbb{E}_{\mathcal{D}_a} \left[ \|\hat{\boldsymbol{a}} - \boldsymbol{a}\|_2^2 \right]$ and where $\lambda = |\mathcal{D}_a|/|\mathcal{D}|$.

**Proposition 4.4** (Action prediction can denoise). *Following the conditions in Proposition 4.3 and assuming $\mathbb{E}[\boldsymbol{q}^T \boldsymbol{\epsilon}] = 0$ and $d_z \geq d_a$, optimizing $\mathcal{L}_a$ biases the encoder parameter perpendicular to the noise. For an artificial case where $\lambda \to +\infty$, we obtain perfect LAM with $D\boldsymbol{\epsilon} = \mathbf{0}$ and $B\boldsymbol{z} = \boldsymbol{q}$.*

We provide the proof in Appendix C.3. Note that we assume that $\boldsymbol{\epsilon}^T X = 0$, meaning that the action effect $\boldsymbol{q} = X\boldsymbol{a}$ and the noise $\boldsymbol{\epsilon}$ are independent. For example, this would hold in the case of table-top manipulation with passers-by occasionally walking by in the background as the source of $\boldsymbol{\epsilon}$ (cf. Case 3 in Section 4.1). The conclusion $D \perp \boldsymbol{\epsilon}$ indicates that noise will not enter the latent (noting that $C = -D$ in this case).

Proposition 4.4 indicates that auxiliary action prediction helps linear LAM to reduce the noise in its latents. Figure 4 (right) present our simulation, we apply different levels of 'other agents' noise (case 3 in Section 4.1) with the same settings in Figure 3 (right). Unlike Figure 3 (right), where $\sigma_{exo} = 2.0$ caused linear LAM to fail, now 1% of action labels in auxiliary action prediction leads to successful learning. The simulation results indicate that only a small proportion of action labels are needed to encourage the latent to focus on encoding real actions, and not noise.

## 5 Beyond Linear LAM Experiments

To test whether the theoretical insights from our linear LAM analysis hold in more realistic settings, we ran empirical experiments using a more realistic LAM setup, on a carefully designed synthetic dataset, which allows us to carefully measures proxies for LLO. We use small 4×4 images as input (instead of vectors), non-linear CNNs in IDM and FDM (instead of linear layers), and vector quantization (instead of using a reduced linear dimension as a bottleneck). We will show that the main conclusions in our paper still hold in this more complex setting.

**Dataset.** We designed a $4 \times 4$ grid-world style synthetic dataset. The top $3 \times 4$ grid of the observation contains a square (intensity=1.0) that can be controlled with five actions (up, down, left, right, and stay still). The bottom $1 \times 4$ grid of the observation contains random Bernoulli noise (with prob 0.5). An intensity parameter controls the noise magnitude (none=0.0, low=1.0, high=2.0).

**Policy.** By default, we use a uniform policy, where each action is equally probable. For one experiment, we also use a correlated policy, where state and action are correlated. With 95% prob, the action moves the square on a fixed snaking pattern through the grid, and with 5% chance a random action is selected.

**Model.** For the IDM, we use a small CNN to encode $\boldsymbol{o}$ and $\boldsymbol{o}'$, followed by a VQ bottleneck with codebook size of 5 outputting the latent. Finally, for the FDM, a separate UNet takes the latent and previous observation $\boldsymbol{o}$ to output the predicted $\hat{\boldsymbol{o}}'$. When predicting actions, codes are preassigned to actions, and latents are trained to minimize L2 distance to their true action code. For data augmentation, we shift the $4 \times 4$ image left/right for one grid with periodic padding. Models

| Setting | Controllable loss ($\downarrow$) | Stochastic loss ($\uparrow$) |
|---|---|---|
| No noise | $\mathbf{0.624 \pm 0.087}$ | – |
| Low noise | $0.781 \pm 0.079$ | $0.739 \pm 0.047$ |
| High noise | $1.046 \pm 0.017$ | $0.607 \pm 0.021$ |
| Uniform policy | $\mathbf{0.781 \pm 0.079}$ | $\mathbf{0.739 \pm 0.047}$ |
| Correlated policy | $1.997 \pm 0.022$ | $0.599 \pm 0.036$ |
| No data augmentation | $0.781 \pm 0.079$ | $0.739 \pm 0.047$ |
| Data augmentation | $\mathbf{0.415 \pm 0.020}$ | $\mathbf{0.898 \pm 0.049}$ |
| No action prediction | $0.781 \pm 0.079$ | $0.739 \pm 0.047$ |
| 1% action prediction | $\mathbf{0.295 \pm 0.011}$ | $\mathbf{0.986 \pm 0.002}$ |

Table 1: Ablations of performance of practical LAM under different settings. We present the mean and standard errors of controllable loss (measures prediction loss of true action from latent, lower means more action information) and stochastic loss (measures prediction loss of noise from latents, higher means less noise information) over five seeds. Each group corresponds to the analysis of one sub-section in Section 4 and we **bold** the losses of the better variant in each group.

were trained for 16k updates with the Adam optimizer. Unless specified, we use low stochastic noise, no action prediction, no data augmentation, and five codebook vectors.

**Evaluation.** Not being linear, it is not possible to measure the mutual information between latent and quantities of interest exactly. Instead, since by design the observation separates the controllable (top half) from stochastic (bottom half) changes, we measure the reconstruction loss on the relevant portion of the observation to assess how what information the latent has captured following training. A lower controllable loss means the latent contains more information about the action, and lower stochastic loss means more noise has been captured. Similar to LLO, this is normalized by the variance of the signal. Note there is unfortunately no straightforward way to measure the information about the observation $o$ in the latent for this non-linear case.

**Results.** We present experiment results containing controllable loss and stochastic loss in Table 1. Firstly, we train vanilla LAM on different noise levels. Results show that as noise is increased from none to high, the action information in the latent decreases (increasing controllable loss). At the same time, the information about the stochastic noise increases (decreasing stochastic loss). This is consistent with our linear LAM theory in Section 4.1, that latents encode whatever leads to most variance. Secondly, we see that switching from a uniform to a correlated policy reduces the information about actions in the latents, while increasing the information about stochastic noise. This is predicted by our linear LAM theory in Section 4.2, that deterministic data collection policy can lead to degenerated performance. Thirdly, we see that data augmentation improves LAM learning. This is consistent with the conclusion in Section 4.3, that data augmentation can help LAM learning. Finally, we show that incorporating 1% of actions labels into the training process improves over vanilla LAM by increasing information about actions, and reducing information about noise. This is consistent with the conclusion in Section 4.4, that predicting true actions improves LAM learning.

## 6   Conclusion

Latent action models (LAMs) are increasingly used in the pre-training phase of embodied AI models. However, they have so far been intuitively and empirically motivated, without thorough analysis into their learnability. This paper bridges this gap by proposing to study LAMs through a simple model capturing the essence of LAM training while remaining mathematically tractable. Through our analysis of linear LAM, we made several observations, including showing that under certain assumptions, it corresponds precisely to PCA. In several noise settings, linear LAM leads to recovery of the true action – this supports the use of real-world LAM in low noise settings, as well as the application of preprocessing steps to reduce noise in a dataset. However, we also showed that when the main cause of variation between consecutive observations is *not* the controllable signal, LAM latents will capture noise rather than action information. We observed a further danger so far undocumented in the literature – a lack of randomization in a data collection policy can harm the latents learned by LAM. Finally, we provided analytical results justifying techniques emerging in the LAM literature to improve the quality of latents learned – data augmentation and auxiliary action prediction.

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

# A  From practical LAM to linear LAM.

Our goal in designing linear LAM was to preserve the key features of LAMs in practice, while making the resulting model as simple as possible. Here we remark on similarities and differences.

- *Function approximation.* Linear LAM uses linear layers, while practical LAM uses deep neural networks. However, linear LAM is compatible with input vectors processed by other non-linear pre-trained image encoders.

- *Bottleneck.* Both models have an information bottleneck between in the IDM but this is implemented in different ways. Practical LAMs usually adopt vector quantization Van Den Oord et al. (2017), while linear LAM uses a low continuous dimension $d_z \ll d_o$.

- *Additive changes.* We formulate changes between observations as additions of controllable changes and exogenous noise. This additive structure provides the simplest combination of two elements which we believe are a primary concern of whether LAM's latents actually represent actions.

- *Noise.* While our model constrains the noise as additive, it does not make further assumptions. We later analyze realistic cases corresponding to real world scenarios, such as when action and noise are correlated, and action and observations are correlated.

# B  Simulation Set-ups

We adopt the following setting unless otherwise stated. We provide the source code in the appendix.

- *Observations.* Observations are sampled from the standard normal distribution, $o \sim \mathcal{N}(\mathbf{0}, I)$ with $I$ as the identity matrix. Note that $\mathbb{V}\mathrm{ar}(o) = 1$ (i.e., each element in $o$ has unit variance). We set the dimension of the observations $d_o = 128$.

- *Actions.* We use continuous actions also sampled from a standard normal distribution with dimensionality $d_a$, so $a \sim \mathcal{N}(\mathbf{0}, I)$. Note that $\mathbb{V}\mathrm{ar}(a) = 1$. We set $d_a = 8$ unless otherwise stated.

- *Controllable changes.* The action effect matrix $X$ that maps the action $a$ to the controllable changes $q$, is chosen as a random orthogonal matrix (using QR decomposition), subsequently normalized to ensure $\mathbb{V}\mathrm{ar}(q) = 1$.

- *i.i.d. noise.* In our simulation we use isotropic Gaussian $\epsilon \sim \mathcal{N}(\mathbf{0}, \sigma_\epsilon^2 I)$ with variance $\sigma_{iid}^2$ as the i.i.d. noise. For this noise, $\mathbb{V}\mathrm{ar}(\epsilon) = \sigma_{iid}^2$.

- *Exogenous noise.* We also consider the noise induced by other agents $\epsilon = Yb = \sigma_{exo}Y_0 b$ where $b$ is other agents' action, $Y$ is the action effect matrix of other agents, and $Y_0$ is the normalized matrix of $Y_0$. Similarly, $Y_0$ is chosen as a random orthogonal matrix using QR decomposition to ensure that $\mathbb{V}\mathrm{ar}(\epsilon) = \sigma_{exo}^2$.

- *Optimization.* We implement our system in PyTorch, optimizing trainable parameters via stochastic gradient descent with the Adam optimizer with batch size 128. We use the default learning rate and run for 4,000 steps to ensure convergence.

- *Evaluation.* Our use the quantity defined in (6) as the default evaluation metric. For the experiments that do not involve noise, we set normalized MSE for noise to 1.

- *Data augmentation.* Data augmentation is a trick that may improve the learnability of LAM mentioned in several previous papers (Chen et al., 2024b; Sun et al., 2024). We implement data augmentation by adding a Gaussian noise $\kappa \sim \mathcal{N}(\mathbf{0}, |\kappa|^2 I)$ for linear LAM. Note that $\mathbb{V}\mathrm{ar}(\kappa) = |\kappa|^2$. By default data augmentation is turned off, except for Figure 4 (middle and right).

- *Action prediction.* Action prediction is another trick proposed in the previous paper (Nikulin et al., 2025). We implement action prediction by predicting the true action label based on the latent with a learnable linear transformation for a small proportion of the data samples. We denote the ratio of the samples that we access their action labels as $\lambda$. We find that setting $\lambda = 1\%$ is enough. By default action prediction is turned off, except for Figure 4 (right).

## C  Proofs

### C.1  Proof of Proposition 4.1

*Proof.* Expand the loss function in (4) with the definitions of $\hat{o}'$ (3) and $o'$ ($o' = o + q + \epsilon$), then rearrange (ignoring the expectation outside the RHS).

$$\mathcal{L} = \|o' - \hat{o}'\|_2^2 \tag{12}$$

$$= \|o' - (Ao + BCo + BDo')\|_2^2 \tag{13}$$

$$= \|(o + q + \epsilon) - (Ao + BCo + BD(o + q + \epsilon))\|_2^2 \tag{14}$$

$$= \|(A + BC + BD - I)o + (BD - I)(q + \epsilon)\|_2^2 \tag{15}$$

$$= \|(A + BC + BD - I)o\|_2^2 + \|(BD - I)(q + \epsilon)\|_2^2$$
$$+ 2o^T(A + BC + BD - I)^T(BD - I)(q + \epsilon) \tag{16}$$

$$= \|(A + BC + BD - I)o\|_2^2 + \|(BD - I)(q + \epsilon)\|_2^2$$
$$+ 2\operatorname{Tr}\left((A + BC + BD - I)o(q + \epsilon)^T(BD - I)\right) \tag{17}$$

Recall that this loss $\mathcal{L}$ is found within an expectation in (4). By assumption $\mathbb{E}[o(q + \epsilon)^T] = \mathbf{0}$ and the final term can be ignored (since both expectation and trace are additive).

$$\mathcal{L} = \mathbb{E}\left[\|(A + BC + BD - I)o\|_2^2\right] + \mathbb{E}\left[\|(BD - I)(q + \epsilon)\|_2^2\right] \tag{18}$$

Regarding the first term, note that $A$ is full rank $d_o$ while $BC$ and $BD$ are of rank $d_z$. Since $A$ only appears in this term and it is of greater or equal rank than $BD + BD$, it's optimal value is $A = I - B(C + D)$, setting the first term is zero. Hence we are left with the middle term.

$$\mathcal{L} = \|(BD - I)(q + \epsilon)\|_2^2 \tag{19}$$

$\square$

Note that, under the condition of Proposition 4.1, $(A, B, C, D)$ are over-parameterized. When we adopting data augmentation in Proposition 4.3, we obtain $A = I$ and $C + D = 0$, which refines how the condition $A = I - B(C + D)$ is satisfied.

### C.2  Proof of Proposition 4.3

We start by expanding the loss defined in (1) with the data augmentation scheme, and unpack (ignoring the expectation outside the RHS).

$$\mathcal{L} = \|\operatorname{Aug}_2[o'] - \hat{o}'\|_2^2 \tag{20}$$

$$= \|o' + \kappa_2 - (A(o + \kappa_2) + B(C(o + \kappa_1) + D(o' + \kappa_1)))\|_2^2 \tag{21}$$

$$= \|(o + q + \epsilon) + \kappa_2 - (A(o + \kappa_2) + B(C(o + \kappa_1) + D(o + q + \epsilon + \kappa_1)))\|_2^2 \tag{22}$$

$$= \|(A + BC + BD - I)o + (BD - I)(q + \epsilon) + (BC + BD)\kappa_1 + (I - A)\kappa_2\|_2^2 \tag{23}$$

$$= \|(BD - I)(q + \epsilon)\|_2^2 + \|(A + BC + BD - I)o\|_2^2$$
$$+ \|B(C + D)\kappa_1\|_2^2 + \|(I - A)\kappa_2\|_2^2 \tag{24}$$

Where the last line follows since $\kappa_1$ and $\kappa_2$ are sampled independently of all other terms and $\mathbb{E}[o^T(q + \epsilon)] = 0$. Hence, we are left with the vanilla linear LAM loss in (15) plus two additional terms.

Minimizing this function can be achieved by exactly setting $A = I$ and $C = -D$, which zeros the last three terms simultaneously. In this case, the optimization problem again reduces to PCA $\|(BD - I)(q + \epsilon)\|_2^2$.

**Discussion.** In the correlated case $\mathbb{E}[o^T(q + \epsilon)] \neq 0$, when the third term (the cross term) in (17) cannot be ignored, nevertheless there is encouragement to reach $A = I$ and $C = -D$.

## C.3 Proof of Proposition 4.4

Based on the conclusion in Proposition 4.3, optimizing for the first term in $\mathcal{L}_a$ results in $A = I$ and $C = -D$. We consider the case where $\mathcal{D}$ and $\mathcal{D}_a$ come from the same distribution and the availability of action labels are uniformly random. Therefore, we can ignore the expectation outside the RHS (for simplicity) and re-write the loss as,

$$\mathcal{L}_a = \|(BD - I)(\boldsymbol{q} + \boldsymbol{\epsilon})\|_2^2 + \lambda\|ED(\boldsymbol{q} + \boldsymbol{\epsilon}) - \boldsymbol{a}\|_2^2 \tag{25}$$

$$= \|(BD - I)\boldsymbol{q}\|_2^2 + \|(BD - I)\boldsymbol{\epsilon}\|_2^2 + \lambda\|ED\boldsymbol{q} - \boldsymbol{a}\|_2^2 + \lambda\|ED\boldsymbol{\epsilon}\|_2^2 \tag{26}$$

$$= \|(BDX - X)\boldsymbol{a}\|_2^2 + \|(BD - I)\boldsymbol{\epsilon}\|_2^2 + \lambda\|(EDX - I_{d_a})\boldsymbol{a}\|_2^2 + \lambda\|ED\boldsymbol{\epsilon}\|_2^2 \tag{27}$$

The second line leverage the fact that $\mathbb{E}[\boldsymbol{q}^T\boldsymbol{\epsilon}] = 0$. We decompose $X$ as $X = U\Sigma V^T$ with $U \in \mathbb{R}^{d_o \times d_a}$ (which spans the column space of $X$), $\Sigma \in \mathbb{R}^{d_a \times d_a}$, and $V \in \mathbb{R}^{d_a \times d_a}$.

We will show that, when $\lambda \to +\infty$, $D = \begin{bmatrix} U^T \\ 0 \end{bmatrix}$, $B = [U, *]$, and $E = [V\Sigma^{-1}, *]$ minimizes $\mathcal{L}_a$ where $*$ indicates arbitrary entries and $*$ vanishes when $d_z = d_a$.

First, this solution zeros the last two terms. For the last term, $D\boldsymbol{\epsilon} = \begin{bmatrix} U^T \\ 0 \end{bmatrix} \boldsymbol{\epsilon} = 0$ since $\boldsymbol{\epsilon}^T X = 0$ and $U$ spans the column space of $X$. For the third term, $EDX = [V\Sigma^{-1}, *] \begin{bmatrix} U^T \\ 0 \end{bmatrix} U\Sigma V^T = I_{d_a}$.

Then, since we have determined $D$, the second term becomes $\|(BD - I)\boldsymbol{\epsilon}\|_2^2 = \|\boldsymbol{\epsilon}\|_2^2$. We can choose $B = [U, *]$ to zero the first term since $BDX = [U, *] \begin{bmatrix} U^T \\ 0 \end{bmatrix} X = X$. In this way, we obtain the minimal loss $\mathcal{L}_a^* = \|\boldsymbol{\epsilon}\|_2^2$. We can also verify that $B\boldsymbol{z} = B(C\boldsymbol{o} + D\boldsymbol{o}') = BD\boldsymbol{q} = UU^T\boldsymbol{q} = \boldsymbol{q}$.

Though the $\lambda \to +\infty$ setting is artificial, the analysis show that a positive $\lambda$ will bias the encoder $D$ to capture less about the noise $\boldsymbol{\epsilon}$ and more about the signal $\boldsymbol{q}$ (or $\boldsymbol{a}$).

