# OpenReview forum: "What Do Latent Action Models Actually Learn?"
_NeurIPS.cc/2025/Conference — NeurIPS 2025 poster_

### Official Review · Reviewer_ojQq · 2025-06-30

**Clarity:** 2
**Significance:** 2
**Originality:** 3
**Rating:** 4
**Confidence:** 4

**Summary:**

The authors introduce a linear abstraction of LAM (linear LAM) that keeps the standard inverse/forward-dynamics architecture but replaces the neural networks with linear maps. This preserves the training signal (reconstruction of observations), while making the model fully analysable. Using the linear form, they deliver theoretical results on the properties of a linear LAM and substantiate them using light-weight simulations that demonstrate the effects of data collection, over-parameterisation, data augmentation and weak supervision. The paper provides an insightful account of the things a linear LAM will learn under different conditions, contributing to the general understanding of the sort of conditions we should create for linear LAMs to learn effectively.

**Questions:**

1. I would appreciate if the authors could improve the writing in the paper; this would be essential for making it more accessible to the readers. (e.g. it is unclear to me what 'practical LAM' means when it is first introduced).

2. L147: It seems that minimising mutual information with an observation should be argued more. Isn't it possible to imagine the case where certain actions are only taken in specific states?

3. L230: The formulation "of other agents" is a bit confusing. Wouldn't this better be phrased as predictable changes that are not being caused by agent's actions?

4. L279: "Therefore, the analysis in this section suggest us to use the data collected by the policies with higher randomness whenever possible." Although the theoretical component is somewhat insightful, this point seems rather trivial at first glance. If the policy is deterministic, a world model would naturally be capable of implicitly modelling the behaviour of the agent?

5. As mentioned in the Strenghts & Weaknesses section, I believe the authors should more clearly address the apparent gap between the presented results and the settings in which LAMs would most likely be applied (complex, high-dimensional settings; large-scale video data). If addressed convincingly, or if the authors can argue their case in the rebuttal, I would be happy to increase my score.

6. IGOR paper claims: "In practice, we found that the video quality has a big impact on the model performance. We exclude low-quality videos characterized by excessive shakiness or rapid camera movement, and apply stabilization techniques to the remaining videos." I may be mistaken, but this slightly contradicts the conclusions of your paper? Could this be initial evidence that your claims do not translate to more complex settings?

**Ethical Concerns:**

["NO or VERY MINOR ethics concerns only"]

**Final Justification:**

I believe the paper has its merits. The theoretical analysis may be useful for future work in LAMs.

My main issue with the paper, lack of experiments showcasing that their results translate to complex and relevant datasets for LAM models, was not fully resolved with the new experiment. Nevertheless, the authors cited relevant work that justifies some of their theoretical results. The authors promised to fix the formating and writing issues.

All in all, I believe it's a borderline accept.

**Limitations:**

The aforementioned gap is recognised, but is not discussed sufficiently, in my opinion.

**Quality:**

3

**Strengths And Weaknesses:**

1. Quality:
- (+) The work is technically sound, providing proofs where necessary.
- (+/-) The theoretical claims about the properties of a linear LAM are well-supported using a number of light-weight experiments. Within the broader scope of LAM usage (mostly large-scale video data), it is unclear whether these claims remain impactful. No thorough investigation into this has been provided.
- (+) The analysis of linear LAM is convincing and looks sufficient.
- (-) The afore-mentioned weakness of the paper remains largely unaddressed in the paper. It would be great to see more justification for how this work may translate to more complex domains.

2. Clarity:
- (-) The writing could be substantially improved. It is often difficult to understand what is being said in the paper.
- (+) The paper is well-organised and follows a good and clear structure.
- (+) The paper contains a sufficient amount of information and details to reproduce the results.
- (-) Figures could be improved for better readability. The y-axis should be labelled.

3. Significance:
- (-) At this point, it is unclear to me whether the analysis presented in the paper may be impactful for the community. My major concern is the vast oversimplification of the underlying model. Although the motivation for this oversimplification is clear (the model becomes analysable), it nevertheless generates a large gap between the results presented in the paper and the actionable significance of these results on the existing LAM methods. Primarily, can we trust these results to be applicable to complex, high-dimensional domains? To be clear, I do not necessarily think that they are not, rather it seems that the paper does little to address this concern in general. As such, it would be great to hear whether the authors believe these concerns could be addressed with more experiments in more complex settings, without taking away from the main claims of the paper.
- (+) The paper does a good job of studying the linear LAM, demonstrating insightful results into its properties.

4. Originality:
- (+/-) While it seems conceivable that the results presented in the paper may be insightful with respect to the existing LAM methods, the point about applicability of these results to the existing LAM models remains unchanged. The authors should provide more details / justifications as to how their analysis may be applicable. What are the limitations? Where, if at all, do the assumptions break down?
- (+) The work presents a first account of theoretical analysis of a linear LAM. The relevant citations are provided.

---

> ### Author Rebuttal · Authors · 2025-07-31
>
> Many thanks for your detailed review. We have aimed to address your concerns, including **a new set of experiments bridging the gap between linear and practical LAM (see the results in the response to the first reviewer)**. We hope you might view this as sufficient to update the paper’s score.
>
> ***[W1] Within the broader scope of LAM usage (mostly large-scale video data), it is unclear whether these claims remain impactful. No thorough investigation into this has been provided. The afore-mentioned weakness of the paper remains largely unaddressed in the paper. It would be great to see more justification for how this work may translate to more complex domains.***
>
> Thank you for this, which we agree was an important point not fully addressed in the initial paper. The gap between linear LAM and LAM usage in practice can be summarized as 1) linear to nonlinear model architecture, 2) continuous to sometimes vector quantized bottleneck; and 3) linear to complex world dynamics in video data (cf. Appendix A). To justify how our conclusions can be translated to complex domains, we conducted an additional set of experiments. Please see the new experiment results in the response to the first reviewer (Reviewer VEZh).
>
> ***[W2] The writing could be substantially improved. It is often difficult to understand what is being said in the paper.***
>
> Apologies for not having had time to thoroughly edit the paper before submission. We commit to doing a full edit in the next version.
>
> ***[W3] Figures could be improved for better readability. The y-axis should be labelled.***
>
> We agree. Currently the y-axis is only labeled in the title, we will revise.
>
> ***[W4] At this point, it is unclear to me whether the analysis presented in the paper may be impactful for the community. My major concern is the vast oversimplification of the underlying model. Although the motivation for this oversimplification is clear (the model becomes analysable), it nevertheless generates a large gap between the results presented in the paper and the actionable significance of these results on the existing LAM methods. Primarily, can we trust these results to be applicable to complex, high-dimensional domains? To be clear, I do not necessarily think that they are not, rather it seems that the paper does little to address this concern in general. As such, it would be great to hear whether the authors believe these concerns could be addressed with more experiments in more complex settings, without taking away from the main claims of the paper.***
>
> Thank you, hopefully our experiments, shared above in W1 response, using nonlinear model architecture, vector quantized bottleneck, and image-based input, will help alleviate this concern.
>
> ***[W5] The authors should provide more details / justifications as to how their analysis may be applicable. What are the limitations? Where, if at all, do the assumptions break down?***
>
> Apart from the new experiment results, the reason why linear LAM would catch the essence of LAM in practice is because it captures the key designs including IDM, FDM, bottleneck and reconstruction loss. The assumption that the dynamics is linear is made to match the model capacity of linear LAM, and we believe this can scale to the scenario where both are nonlinear. The limitation of linear LAM is that it cannot shed light on the LAM research on deep learning techniques (e.g., cross-attention vs self-attention in IDM), empirical data selection (e.g., the best proportion for each real dataset), or optimization (e.g., the optimizer or the learning rate schedule). We argue that the value of our paper is not we can obtain results based on assumptions that may not hold. Instead, our value is that the conclusions drawn from our analysis can be applied to the practice (e.g., using more exploratory data to train LAM) or explain the observations in practice (e.g., the data augmentation technique).
>
> ***[Q1] I would appreciate if the authors could improve the writing in the paper; this would be essential for making it more accessible to the readers. (e.g. it is unclear to me what 'practical LAM' means when it is first introduced).***
>
> See W2 response. We will add a practical LAM definition.
>
> ***[Q2] L147: It seems that minimising mutual information with an observation should be argued more. Isn't it possible to imagine the case where certain actions are only taken in specific states?***
>
> Minimizing mutual information with observation is still a proper objective, even if there are specific states where only certain actions can be taken. In this case, it is true that the mutual information between latent and state cannot be zero since we can infer the state from the latent. However, minimizing this mutual information still helps LAM to discover action representations to be as independent with states as possible so that it can be shared across states. Note that the ultimate objective for LAM is to learn shared action representation with consistent semantics (e.g., the latent could be “moving the arm left if not blocked”).
>
> ***[Q3] L230: The formulation "of other agents" is a bit confusing. Wouldn't this better be phrased as predictable changes that are not being caused by agent's actions?***
>
> We agree this term could be changed to improve clarity. We propose to use use “predictable noise” in our revision – please let us know if this satisfies your concern.
>
> ***[Q4] L279: "Therefore, the analysis in this section suggest us to use the data collected by the policies with higher randomness whenever possible." Although the theoretical component is somewhat insightful, this point seems rather trivial at first glance. If the policy is deterministic, a world model would naturally be capable of implicitly modelling the behavior of the agent?***
>
> Our main observation is that, when the action is easily predicted from the current state, there is less incentive for the latent to encode the action, with the FDM (or a world model as the reviewer mentioned) predicting the next state directly from the current state. Hence, we should prefer a higher entropy policy for the purpose of training a LAM. Like all good insights, as you note, this is intuitive, but we only made this discovery through our analysis, and do not believe related literature has made this observation.
>
> ***[Q5] As mentioned in the Strenghts & Weaknesses section, I believe the authors should more clearly address the apparent gap between the presented results and the settings in which LAMs would most likely be applied (complex, high-dimensional settings; large-scale video data). If addressed convincingly, or if the authors can argue their case in the rebuttal, I would be happy to increase my score.***
>
> In response to the reviewer’s concern about the gap, we have provided additional experiment results in the response to the first reviewer (Reviewer VEZh)..
>
> ***[Q6] IGOR paper claims: "In practice, we found that the video quality has a big impact on the model performance. We exclude low-quality videos characterized by excessive shakiness or rapid camera movement, and apply stabilization techniques to the remaining videos." I may be mistaken, but this slightly contradicts the conclusions of your paper? Could this be initial evidence that your claims do not translate to more complex settings?***
>
> No. Filtering data is an orthogonal technique from data’s perspective. Combining with the information that IGOR also uses data augmentation (advocated by our paper), we conclude that camera shakiness could be an issue and both the data augmentation (to help LAM robust to gentle shakiness) and data filtering (to avoid LAM suffer from excessive shakiness) are needed.

---

> > ### Comment · Reviewer_ojQq · 2025-08-03
> >
> > Hey! Thanks for the response.
> >
> > I appreciate the detailed response. I believe the authors largely addressed my concerns; however, I am still struggling with the most important one.
> >
> > I see you have conducted a small-scale grid world experiment. I have a few questions about that:
> > 1. Is the image 4x4 pixels?
> > 2. You say you separated the image into two; is it then 2x1 for the top half and 2x1 for the bottom half?
> > 3. Can the agent move to the bottom half of the image? The controllable loss measures the reconstruction loss wrt the top half, but what if the agent is in the bottom half?
> > 4. Have you run the experiments over multiple seeds? I understand the time constraint, but these experiments are quite important for the paper.
> >
> > Results:
> > 1. "the action information in the latent decreases" -- hard to tell without error bars?
> > 2. Other results seem more conclusive, though error bars would still be appreciated.
> >
> > I am currently unclear whether this experiment fully addresses my initial concern. The experiment setting seems too simple (e.g., the dynamics of the gridworld). Further, could the authors justify more thoroughly why they think the reconstruction loss is a good proxy for mutual information?
> >
> > Finally, it would be good if the authors could clarify why they think this experiment is sufficient to showcase that the conclusions of the paper are applicable to LAMs applied to complex datasets (e.g., Mujoco with significantly more complex dynamics).

---

> > > ### Author Response · Authors · 2025-08-05
> > >
> > > Thank you for communicating key points you require clarification on. We respond to your queries below and will revise our paper accordingly.
> > >
> > > ***On the experiment setup***
> > >
> > > At the suggestion of the reviewers, the aim of our new experiment was to investigate whether our results hold when the gap from linear to practical LAM is closed. Our experiment closes this gap in at least 3 ways.
> > > 1) Moves from linear models to neural networks.
> > > 2) Moves from a linear dimesion bottleneck, to a VQVAE standard in LAMs.
> > > 3) Moves away from linear dynamics and noise function.
> > > Note that with this significant departure from linear LAM, we have been able to confirm all major intuitions from our theory.
> > >
> > > However, we agree with the reviewer that the dynamics and input space remain far simpler than, say, real-world robotics.
> > > To this, we argue that prior works have already observed gains from some of our insights on real data, through pure experimental study, and could be viewed as already providing empirical evidence of this.
> > > For example [1] use augmentation, and [2] measure the effect of action prediciton.
> > >
> > > We see the contribution of our paper as analyzing and understanding the cause of these modifications in an interperetable and controlled setting. E.g. we provide analysis that data augmentation provably can improve the learning of LAM, albeit under certain assumptions.
> > > Our analysis also brings to light new issues not known to the community (namely the effect of data collection policy).
> > >
> > > [1] Chen et al. *Igor: Image-goal representations...* arXiv:2411.00785
> > >
> > > [2] Nikulin et al. *Latent action learning...* arXiv:2502.00379
> > >
> > > ***Q1 Observation dims***
> > >
> > > That's correct, the input is currently a 4x4 pixel grid. If you feel it is important to the paper, we would be happy to commit to repeating the runs with the same dynamics over a larger grid, or rendering the current 4x4 grid world to a larger size image. We would not expect either to effect our conclusions.
> > >
> > > ***Q2 Observation details***
> > >
> > > Thank you for this question, since we didn't originally have sufficient space to explain.
> > > * The **upper portion** of the observation corresponds to a **(3, 4) grid** that the controllable agent may move within.
> > > * The **lower portion** corresponds to a **(1, 4) grid**, representing (only) stochastic noise.
> > >
> > > ***Q3 Controllable regions***
> > >
> > > The agent cannot move to the bottom half of the image. If it tries to move outside of the (3,4) area, it's position remains constant.
> > >
> > > As you note, this is an important detail, because if this weren't the case our reconstruction metrics for controllable and stochastic would be entangled.
> > >
> > >
> > > ***Q4 Error bars***
> > >
> > > Thank you for encouraging the use of error bars. We have rerun all experiments with 5 random seeds, and below present mean and standard errors. These support all our earlier observations, including "as noise is increased from none to high, the action information in the latent decreases".
> > >
> > > |  | Controllable loss | Stochastic loss |
> > > |--|--|--|
> > > | A) No noise | 0.624 ± 0.087| 0.0 ± 0.0 |
> > > | A) Low noise | 0.781 ± 0.079| 0.739 ± 0.047 |
> > > | A) High noise | 1.046 ± 0.017| 0.607 ± 0.021 |
> > > | B) Uniform policy | 0.781 ± 0.079 | 0.739 ± 0.047 |
> > > | B) Correlated policy | 1.997 ± 0.022 | 0.599 ± 0.036 |
> > > | C) No data augmentation   | 0.781 ± 0.079         | 0.739 ± 0.047   |
> > > | C) Data augmentation      | 0.415 ± 0.020         | 0.898 ± 0.049   |
> > > | D) No action prediction   | 0.781 ± 0.079         | 0.739 ± 0.047   |
> > > | D) 1\% action prediction  | 0.295 ± 0.011         | 0.986 ± 0.002   |
> > >
> > > ***Q5 Why is reconstruction loss a good proxy for mutual information?***
> > >
> > > The unavoidable downside of moving away from our linear model, is that measuring and analyzing certain things becomes challenging.
> > > In fact, since all actions, noise and latent are discrete in our grid-world setting, we originally hoped to be able to compute MI exactly.
> > > However, we found that models do not learn a one-to-one mapping from the latent codes to actions or noise, and thus MI cannot be easily calculated by its definition.
> > > The action appears to get encoded in a more convoluted way, being dependent on the observation (e.g. one could imagine a scheme where the effect of a specific latent changes depending on which row of the grid the agent is in).
> > >
> > > However, we noted that in our LAM system, any information about both action and noise in the output prediction must have also been present in the latent (since there is only one route for this through the model).
> > > Hence, the reconstruction loss over the relevant portion (controllable or stochastic) of the observation gives a kind of upper bound on how much knowledge the latent contains about the action or noise (the bound comes from the fact the decoder may not do a perfect job of applying the action).
> > >
> > > Hence we believe it is a convenient intuitive proxy for MI. If the reviewer can suggest alternative ways to compute MI in this setting, we would be happy to try them out.

---

> > > > ### Comment · Reviewer_ojQq · 2025-08-05
> > > >
> > > > Thank you for putting in the effort during the rebuttal period and your lengthy clarifications.
> > > >
> > > > I still do not think your additional experiment verifies the claim that the results translate to complex applications. Indeed, I find that, in this case, citing previous work was more convincing.
> > > >
> > > > That being said, I still think the paper has its merits and may serve as a useful guide for future work with LAMs. I would encourage the authors to include more realistic experiments for the final version of the paper, if allowed.
> > > >
> > > > At this point, I am willing to increase my score.

---

### Official Review · Reviewer_oyh9 · 2025-06-30

**Clarity:** 3
**Significance:** 3
**Originality:** 3
**Rating:** 4
**Confidence:** 4

**Summary:**

LAMs are becoming a models of choice, after LAPO paper, for pre-training embodied AI models. But a lot is not known about them, such as does learned latent really encode actions or just noise? Authors answer this question and others via linear LAM model that is easy to analyze theorerically. Having two different random variables, controllable changes q and exogenous noise \eps gives possibilty to understand better what is going on in action prediction. Aurthors also experimentally analyze, via a set of simulations, how different conditions affect the action prediction capability.

**Questions:**

- In line 198, authors invert matrix B, but matrix B is not a square matrix. So you actually use pseudoinverse instead?
-Section 4.1 Case 3, you only state that similar analysis of Case 1 can be conducted and what it results to. But I do not see reference to an appendix wheres such an analysis can be found.
- Section 4.1 constains there cases, where Case 1 has no noise, Case 2 has stochastic noise and Case 3 has a systematic noise. Wouldn't it make sense to have fourth case, which has both stochastic and systematic noise?
- Simulation results seem to produced with one seed, as there is stochasticity in the experimental procedure it is natural to ask whether different seeds to lead to different numbers? I am guessing that trends and conclusions would still hold.

**Ethical Concerns:**

["NO or VERY MINOR ethics concerns only"]

**Final Justification:**

Paper gives insight into LAMs by obtaining theoretical insights from the linear LAM. They answered my most critical issue by performing experiments where they show that with classic LAM they will obtain similar conclusions as with the linear LAM theory.

**Limitations:**

Mostly yes, but I feel that authors should discuss more about how their results can carry over to practical LAMs.

**Paper Formatting Concerns:**

-Line 226 "Gaussion" -> "Gaussian"
-Grammatically sentences do not always parse. A bit more work in this respect is needed.

**Quality:**

3

**Strengths And Weaknesses:**

Strengths:
- Theorerical framework to analyze different aspects of LAMs. Analysis is from multiple perspectives, no exogenous noise to adding data augmentation.
- Proofs in the appendices are easy to read and seem to contain all details.
- I find results to be interesting and illuminating.

Weaknesses:
- It is not clear how results shown here carry over to practical LAMs. I think that is anyways the main point and motivation of this work. Text contains references to practical LAM papers in certain places where obtained results are similar to those in previous studies. This is good, but more effort would be needed here to make results practically useful. Appendix A is also too weak in this respect. Real difference is that dynamics model is not linear in practical LAM. About actions being Gaussian is a desing choice by authors.

- How does linear LAM compare empirically to some practical LAM? Considering that paper already has quite a bit of simulation results it would not be a huge leap to add such a study to appendices.

- Some results are just mentioned in the text, but no complete rationale is given in appendices, such as the difference between Eq. (5) and Eq. (6). Can Eq. (6) be interpreted as mutual information?

---

> ### Author Rebuttal · Authors · 2025-07-31
>
> Thanks for your review. We hope **our extra work in providing new experiments that our results do carry over to practical LAMs (see the results in the response to the first reviewer)** may warrant an increase in score.
>
> ***[W1] It is not clear how results shown here carry over to practical LAMs. I think that is anyways the main point and motivation of this work. Text contains references to practical LAM papers in certain places where obtained results are similar to those in previous studies. This is good, but more effort would be needed here to make results practically useful. Appendix A is also too weak in this respect. Real difference is that dynamics model is not linear in practical LAM. About actions being Gaussian is a design choice by authors.***
>
> We agree that many real-world dynamics are not linear. This choice has been made to allow for tractable analysis and be compatible with linear LAM architecture (i.e., linear LAM can capture linear dynamics, and nonlinear LAM can capture more complex dynamics). Linear LAM design has two benefits: 1) Providing theoretical justification for tricks currently found to work well in practice (such as data augmentation and auxiliary action prediction). 2) Produces new insights into details affecting the learning of LAM (namely we noted the importance of data collection policy, which has not been mentioned in prior literature).
>
> To evaluate how linear LAM captures the real-world LAM, we conduct further experiments as follows to validate that the main conclusion we obtained from linear LAM also applies to the setting where the dynamics is not linear and the action is not Gaussian. Please see the new experiment results in the response to the first reviewer (Reviewer VEZh).
>
> ***[W2] How does linear LAM compare empirically to some practical LAM? Considering that paper already has quite a bit of simulation results it would not be a huge leap to add such a study to appendices.***
>
> The experiment results presented in the response to [W1] indicate that our main arguments still hold for practical LAMs.
>
> ***[W3] Some results are just mentioned in the text, but no complete rationale is given in appendices, such as the difference between Eq. (5) and Eq. (6). Can Eq. (6) be interpreted as mutual information?***
>
> Eq. (6) approximates Eq. (5) by exploiting the connection between mutual information and the MSE loss between the target variable and the least squares estimate (LSE) of the target variable.
>
> ***[Q1] In line 198, authors invert matrix B, but matrix B is not a square matrix. So you actually use pseudoinverse instead? -Section 4.1 Case 3, you only state that similar analysis of Case 1 can be conducted and what it results to. But I do not see reference to an appendix wheres such an analysis can be found.***
>
> Yes, we use pseudoinverse in Line 198. For Section 4.1 Case 3, we can define $\bar{X} = [X|Y] \in \mathbb{R}^{d_o \times (d_a + d_b)}$ (column concatenation) and $\bar{a} := [a^T|b^T]^T$ (row concatenation). Consequently, the results in Case 1 about $X$ and $a$ can be applied directly to $\bar{X}$ and $\bar{a}$ in Case 3. We will add the deduction to the revised appendix.
>
> ***[Q2] Section 4.1 contains three cases, where Case 1 has no noise, Case 2 has stochastic noise and Case 3 has a systematic noise. Wouldn't it make sense to have fourth case, which has both stochastic and systematic noise?***
>
> This is a very good question. Stochastic noise increases the reconstruction loss but does not change the semantics of the learned latent. Systematic noise, when intense enough, can change the semantics of the learned latent. It is a bit hard for us to obtain a clear analytical solution when combining them together, but intuitively they will change both the loss and the latent semantics.
>
> ***[Q3] Simulation results seem to produced with one seed, as there is stochasticity in the experimental procedure it is natural to ask whether different seeds to lead to different numbers? I am guessing that trends and conclusions would still hold.***
>
> We find that seed does not influence the results too much. To show this, we run vanilla linear LAM with $d_a = d_z = 8$ with 10 seeds and find that the standard deviation for linear LAM objective (LLO) is only 0.0173 (compared with the range for LLO [-2, +2]).
>
> ***[F1] Line 226 "Gaussion" -> "Gaussian" -Grammatically sentences do not always parse. A bit more work in this respect is needed.***
>
> We will further improve the paper based on the reviewer’s suggestion.

---

> > ### Comment · Reviewer_oyh9 · 2025-08-01
> >
> > I thank authors for their diligent rebuttal. Most of my concerns have been adequately answered. Especially I am happy to see experiments with standard LAM that seem to validate the theoretical results in linear LAM. Validating experimental results with multiple seeds was also particularly useful.
> >
> > About W3, I would like to see reference to a source where this connection is made. I hope that authors will provide full explanation in the appendices for the completeness sake.

---

> > > ### Author Response · Authors · 2025-08-03
> > >
> > > We sincerely thank the reviewer again for the thoughtful and constructive feedback, which has helped us clarify key aspects of our work. We hope that our detailed response below — along with the additional clarification to be included in the appendix — addresses the reviewer’s concerns. We respectfully ask the reviewer to consider updating their score in light of these improvements.
> > >
> > > Regarding W3, we will include a full explanation in the appendix of the revised paper. Here, we provide a brief sketch of the derivation for completeness.
> > >
> > > We asssume $x$ and $y$ are joint Gaussian variables, i.e.,
> > > $$
> > > \left(
> > > \begin{matrix}
> > > x \\
> > > y
> > > \end{matrix}
> > > \right)
> > > \sim
> > > \mathcal{N}
> > > \left(
> > > \left(
> > > \begin{matrix}
> > > \mu_x \\
> > > \mu_y
> > > \end{matrix}
> > > \right),
> > > \left(
> > > \begin{matrix}
> > > \Sigma_{x} & \Sigma_{xy} \\
> > > \Sigma_{yx} & \Sigma_{y}
> > > \end{matrix}
> > > \right)
> > > \right)
> > > $$
> > >
> > > Following [1], we have $y|x \sim {N}(\mu\_{y} +\Sigma\_{yx}\Sigma^{-1}_{x}(x - \mu_x); \Sigma\_{y|x})$ with $\Sigma\_{y|x}:=\Sigma\_{y} - \Sigma\_{yx}\Sigma^{-1}\_{x}\Sigma\_{xy}$, the least squares estimate (LSE) of $y$ is $\hat{y} = \mathbb{E}[y|x] = \mu\_y +\Sigma\_{yx}\Sigma^{-1}\_{x}(x - \mu\_x)$,
> > > and $\mathbb{E}[||\hat{y} - y||^2] = Tr(\Sigma\_{y|x})$.
> > >
> > > Folloing [2], for Gaussian random variables, we have
> > > $$
> > > I(x;y) = H(y) - H(y|x) = \frac{1}{2} \log ((2\pi e)^n |\Sigma_{y}|) - \frac{1}{2} \log ((2\pi e)^n |\Sigma_{y|x}|) = - \frac{1}{2} \log\left( \frac{|\Sigma_{y|x}|}{|\Sigma_{y}|} \right)
> > > $$
> > > where $n$ is the dimension of $y$, and $|\cdot|$ indicate the determinant.
> > >
> > > To approximate the mutual information in a more computationally efficient way, we propose replacing the determinant of the covariance matrix with its trace. This approximation is motivated by the fact that both quantities reflect a form of “total uncertainty,” but while the determinant captures volume (geometric mean of eigenvalues), the trace captures total variance (sum of eigenvalues). Under certain isotropic or low-rank assumptions, this approximation can still preserve relative scale.
> > >
> > > Thus, we approximate
> > > $$
> > > I(x;y) \approx - \frac{1}{2} \log\left( \frac{\mathbb{E}[||\hat{y} - y||^2]}{Var[y]} \right)
> > > $$
> > > where $Var[y]:=Tr(\Sigma_y)$ denotes the total variance of $y$.
> > >
> > > In the scalar case (i.e., when $x$ and $y$ are real-valued random variables), this approximation becomes exact.
> > >
> > > [1] Christopher Bishop "Pattern Recognition and Machine Learning" Section 1.5.5, 2.3
> > >
> > > [2] https://en.wikipedia.org/wiki/Multivariate_normal_distribution

---

> > > > ### Comment · Reviewer_oyh9 · 2025-08-03
> > > >
> > > > Thanks a lot for the clarification. It is really good that you will add the full derivation to the appendix, even though this result is quite known, to a new researcher in the field it might not be.
> > > >
> > > > I am satisfied with the answers provided by the authors and I am willing to raise my score.

---

### Official Review · Reviewer_EmH8 · 2025-07-01

**Clarity:** 3
**Significance:** 3
**Originality:** 3
**Rating:** 4
**Confidence:** 3

**Summary:**

This paper presents a theoretical analysis of a linear latent action model, offering several insights into its fundamental characteristics and the impact of various data manipulation strategies. In particular, the analysis is based on a bottleneck structure and focuses on the relationship to controllable changes, noise, and observations. The theoretical findings are supported by numerical simulations.

**Questions:**

1. Is the analysis applicable when observations consist of multiple frames? Intuitively, capturing actions based on only two adjacent frames seems limiting.
2. All analysis relies on the additive transition function in Line 139. Is this assumption valid in practice?
3.  I understand in the ideal case when $z$ contains no information about the noise and observation, the LLO value is 2 in Line162, but why this is the optimal value of LLO.
4. Some typos. Line 195, should be $A=(1-\alpha)o$; Line 323, missing $\lambda$ in the objective.

**Ethical Concerns:**

["NO or VERY MINOR ethics concerns only"]

**Final Justification:**

Though the experiments are too simple to capture the complexity of practical applications, the paper presents valuable insights from theoretical perspective, and lays a solid foundation for future research. I will keep my original score, and would like to support this paper.

**Limitations:**

yes

**Quality:**

3

**Strengths And Weaknesses:**

Strength:
1. The paper is clearly written and well-organized.
2. It provides new proofs and theoretical analysis for latent action models, with findings consistent with common practices in the literature.
3. The analysis helps bridge the gap between theory and practice, offering valuable insights for future work.


Weaknesses:
1. The paper is mathematically heavy, making it challenging to fully grasp the details.
2. The analysis is based on simplified linear models with strong assumptions(Line 170), limiting the generalizability and practical applicability of the results to more complex or real-world models.
3. The paper validates its findings solely through numerical simulations, lacking empirical evaluations on real-world datasets to support the theoretical claims.

---

> ### Author Rebuttal · Authors · 2025-07-31
>
> Thank you for taking the time to review our paper. We hope our responses, and **a new set of experiments addressing the reviewers' concern about the gap from linear LAM to real (see the results in the response to the first reviewer)** might justify an uplift in score.
>
> ***[W1] The paper is mathematically heavy.***
>
> Thanks for this comment, since we would like to make the paper accessible to many readers. If the reviewer could mention any specific areas they thought were tricky, we’d be happy to add more intuitive explanations.
>
> ***[W2] The analysis is based on simplified linear model with assumptions (Line 170)***
>
> We have studied the effect of this assumption carefully. Only the analysis in Section 4.1 is based on the assumption in Line 170 (noise and action are uncorrelated with observation). This assumption leads to perfect learnability of LAM. We later relax this assumption in Section 4.2 to show that stochastic data collection policy matters in reducing the correlation between action and observation. Also note that the noise correlated with the observation can be regarded as the signal.
>
> ***[W3] The paper lacks empirical evaluations on real-world datasets***
>
> Thanks for this comment. At the request of the review, to provide evidence on whether our conclusions still hold on real-world datasets which are mainly image-based, we conducted a new set of experiments aimed at bridging the gap between linear LAM and real-world LAM. Please see the new experiment results in the response to the first reviewer (Reviewer VEZh).
>
> ***[Q1] Is the analysis applicable when observations consist of multiple frames? Intuitively, capturing actions based on only two adjacent frames seems limiting.***
>
> Our analysis still applies when we simply redefine the vector observation $\bf{o}$ to contain the information about the history, corresponding to the setting in Genie and LAPO. For example, the learned latent can still capture $\bf{q} + \bf{\epsilon}$ in the idea case (cf. Proposition 4.1). The only difference is that the semantics of $\bf{q} + \bf{\epsilon}$ may be different. Intuitively, the latent may capture the “acceleration” instead of “velocity” when an object moves across multiple frames.
>
> Moreover, even Genie and LAPO still aim at capturing movement between two frames (with additional input about the history). We find this may not be very limiting. Consider another option that extracts a latent to encode the change across multiple frames instead of extracting a latent between each two frames. However, we find that this formulation is not common in practice, maybe because such multi-frame movement can be easily represented as the concatenation of compact representations of two-frame movements or just the latent of two skip frames.
>
> ***[Q2] All analysis relies on the additive transition function in Line 139. Is this assumption valid in practice?***
>
> This linear assumption is a simplification required for tractability. Since our linear LAM architecture is not as complex as practical LAMs, it can only handle additive transition dynamics. We believe that when LAM is more capable (as in practice), it can handle more complex dynamics. Our setting (linear LAM + linear dynamics) is a good approximation of the real case (nonlinear LAM + complex dynamics). Our new set of experiments give evidence that our insights hold in settings when dynamics are not linear.
>
> ***[Q3] I understand in the ideal case when z contains no information about the noise and observation, the LLO value is 2 in Line162, but why this is the optimal value of LLO.***
>
> For a random variable $x$, $||\hat{x} – x||^2 / \text{Var}(x)$ is no smaller than 0 due to the non-negativeness of norm. It equals 0 when $\hat{x}$ is a perfect prediction of $x$. Another notable condition is that $\hat{x}$ is trained to predict $x$ to minimize the MSE loss appearing in the numerator. Since the worst prediction for $\hat{x}$ would be predicting the mean, this term will be no larger than 1. We will make it clearer in our revision.
>
> ***[Q4] Some typos. Line 195, should be A=(1-alpha)o ; Line 323, missing lambda in the objective.***
>
> Thank you, we will fix the typo in Line 323. After checking we believe Line 195 is correct as stands, it should be $A=(1-\alpha) I$ where $I$ is the identity matrix, and thus $A \bf{o} = (1-\alpha) \bf{o}$ which can be used to cancel out the information $\alpha \bf{o}$ mixed in the latent.

---

> > ### Comment · Reviewer_EmH8 · 2025-08-04
> >
> > I thank the authors for their rebuttal. Most of my concerns have been adequately addressed.
> >
> > While I appreciate the inclusion of experiments in real-world settings, I still find the example provided to be too simple to fully capture the complexity of practical applications.
> >
> >  Nonetheless, as a theoretical contribution, the paper presents valuable insights and lays a solid foundation for future research.

---

### Official Review · Reviewer_VEZh · 2025-07-03

**Clarity:** 3
**Significance:** 3
**Originality:** 3
**Rating:** 4
**Confidence:** 3

**Summary:**

This paper proposes a simplified version of latent action models (LAM), called linear LAM, and develops a mathematical theory around it, supported by synthetic experiments. Using this simplified framework, the authors justify the standard training objective for LAM and highlight the risks of over-parameterization. They also theoretically show that correlations between observations and actions can degrade performance. Finally, the paper provides theoretical explanations for how two practical techniques—data augmentation and semi-supervised learning with a small amount of labeled data—can help mitigate the effects of over-parameterization and exogenous noise.

**Questions:**

How well does the simplified linear LAM model capture the behavior of real-world latent action models? Could you discuss the gap between this abstraction and practical architectures, and its impact on the applicability of your theoretical results?

**Ethical Concerns:**

["NO or VERY MINOR ethics concerns only"]

**Limitations:**

I do not see any major ethical or societal concerns raised by this work. The paper acknowledges some limitations.

**Paper Formatting Concerns:**

I did not notice any.

**Quality:**

3

**Strengths And Weaknesses:**

**Strengths**:
- To the best of my knowledge, this is one of the first works to provide a theoretical analysis of the learnability of latent action models (LAM).
- The mathematical derivation is thorough and complemented by intuitive explanations. The paper discusses a range of scenarios that help clarify the theoretical insights.
- The synthetic experiments are well designed, align closely with the theoretical results, and are easy to interpret.
- Although the analysis is conducted in a simplified setting, it may offer useful insights into training behaviors and challenges that arise in practical applications of LAMs, such as imitation learning or offline reinforcement learning.

**Weaknesses**:
- While the simplified linear LAM enables a more tractable analysis, it is unclear how well this abstraction reflects the behavior of LAMs used in practice. It would strengthen the paper if the authors could discuss the gap between the simplified model and real-world architectures in more detail.
- There are minor typos, such as using “a” instead of “an” in some cases. A careful proofreading pass is recommended.
- Results from Section 4.3 are used earlier in Section 4.1. While this is noted in the text, clearer referencing (e.g., citing the relevant proposition) would reduce reader confusion.

---

> ### Author Rebuttal · Authors · 2025-07-31
>
> We thank the reviewer for their valuable feedback. We are grateful that the reviewer acknowledged our paper as “one of the first works to provide a theoretical analysis of the learnability of LAM” with “thorough mathematical derivation” and “intuitive explanations”. We address the concerns as below, and hope these might be worthy of an upgrade in score.
>
> ***[Simplified-to-real gap] Discuss the gap between simplified model and real-world architecture. How well does the simplified linear LAM model capture the behavior of real-world latent action models? Could you discuss the gap between this abstraction and practical architectures, and its impact on the applicability of your theoretical results?***
>
> Thank you for this important point – in order to better understand how linear LAM captures the real-world LAM, we have conducted further experiments to validate that the main conclusions we obtained from linear LAM also applies to the setting that bridges the three gaps between linear and practical LAM: 1) linear to nonlinear model architecture, 2) continuous to sometimes vector quantized bottleneck; and 3) linear to complex world dynamics (also see Appendix A). We present the new experiment results in the end.
>
> ***[Paper writing] There are minor typos and the referencing issue.***
>
> Apologies for allowing these to pass in the first version. We have fixed these issues in the next version.
>
> ---
>
> # New Experiment Results
>
> We provide a set of experiments using practical LAM to understand how our insights from linear LAM analysis transfer to real settings. Specifically, we use image as the input, non-linear CNNs in IDM and FDM, and vector quantization. We show that the main conclusions in our paper still hold in this complex setting.
>
> **Dataset**
>
> We designed a 4x4 grid-world style synthetic dataset. The top half of the observation contains a square (intensity=1.0) that can be controlled with 5 actions (up, down, left, right, and stay still). The bottom half of the observation contains random Bernoulli noise (with prob 0.5). An intensity parameter controls the noise magnitude (none=0.0, low=1.0, high=2.0).
>
> **Policy**
>
> By default, we use a uniform policy, where each action is equally probable. For one experiment, we also use a correlated policy, where state and action are correlated. With 95% prob, the action moves the square on a fixed snaking pattern through the grid, and with 5% chance a random action is selected.
>
> **Model**
>
> For the IDM, we use a small CNN to process $o$ and $o’$, followed by a VQ bottleneck with codebook size of 5 outputting the latent. Finally, for the FDM, a separate UNet takes the latent and previous observation $o$ to output the predicted $\hat{o}’$. When predicting actions, codes are preassigned to actions, and latents are trained to minimize L2 distance to their true action code. For data augmentation, we shift the 4x4 image left/right for one grid with periodic padding. Models were trained for 16k updates with the Adam optimizer.
>
> **Default setting**
>
> Unless specified, we use low stochastic noise, no action prediction, no data augmentation, and 5 codebook vectors.
>
> **Evaluation**
>
> Not being linear, it is not possible to measure the mutual information between latent and quantities of interest exactly. Instead, since by design the observation separates the controllable (top half) from stochastic (bottom half) changes, we measure the reconstruction loss on the relevant portion of the observation to assess how what information the latent has captured following training. A lower controllable loss means the latent contains more information about the action, and lower stochastic loss means more noise has been captured. Similar to LLO, this is normalized by the variance of the signal. Note there is unfortunately no straightforward way to measure the information about the observation $o$ in the latent for this non-linear case.
>
> **Results**
>
> Firstly, we train vanilla LAM on different noise levels. Results show that as noise is increased from none to high, the action information in the latent decreases (increasing controllable loss). At the same time, the information about the stochastic noise increases (decreasing stochastic loss). **This is consistent with our linear LAM theory in Section 4.1, that latents encode whatever leads to most variance.**
>
> |  | Controllable loss | Stochastic loss |
> |--|--|--|
> |No noise |  0.520   | 0.000  |
> |Low noise |  0.951 | 0.694 |
> |High noise |  1.013  | 0.539   |
>
> Secondly, we see that switching from a uniform to a correlated policy reduces the information about actions in the latents, while increasing the information about stochastic noise. **This is predicted by our linear LAM theory in Section 4.2, that deterministic data collection policy can lead to degenerated performance.**
>
> | | Controllable loss | Stochastic loss |
> |--|--|--|
> | Uniform policy  |  0.951 | 0.694 |
> | Correlated policy |  2.022 | 0.579 |
>
> Thirdly, we see that using data augmentation can improve LAM learning. **This is consistent with the conclusion in Section 4.3, that data augmentation can help LAM learning.**
>
> | | Controllable loss | Stochastic loss |
> |--|--|--|
> | No data augmentation |  0.951 | 0.694 |
> | Data augmentation | 0.450 | 0.891 |
>
> Finally, we show that incorporating 1% of actions labels into the training process improves over vanilla LAM by increasing information about actions, and reducing information about noise. **This is consistent with the conclusion in Section 4.4, that predicting true action can improve LAM learning.**
>
> | | Controllable loss | Stochastic loss |
> |--|--|--|
> | No action prediction |  0.951 | 0.694 |
> | 1% action prediction |  0.317 | 0.980 |

---

> > ### Comment · Reviewer_VEZh · 2025-08-04
> >
> > I thank the authors for addressing my questions and especially appreciate the addition of new experimental results. The updates strengthen the paper and provide valuable insights.

---

### Note · Authors · 2025-08-13

We sincerely thank the reviewers and area chair for their service in reviewing our paper – a process which has helped materially improve the quality and clarity of our contributions.

All reviewers (VEZh, EmH8, oyh9, ojQq) acknowledged that this paper provides one of the first theoretical analyses on the learnability of latent action models (LAMs), with thorough mathematical derivations, intuitive explanations, and well-designed synthetic experiments. They valued the paper’s insights into factors influencing LAM learning and noted its potential to guide future research.

Based on reviewers’ constructive feedback, we made several notable additions and clarifications through the rebuttal process:
* [VEZh, EmH8, oyh9, ojQq] We conducted new experiments with a practical form of LAM (nonlinear CNNs, vector-quantized bottleneck, non-linear dynamics). These supported all key findings made through our analysis of linear LAM.
* [VEZh] We provided detailed discussion of the gap between linear and real-world LAMs, and clarified applicability of our theoretical results.
* [EmH8] We explained applicability to multi-frame observations, clarified the additive transition assumption, and expanded derivations (e.g., LLO optimal value) to improve accessibility.
* [oyh9] We clarified connections between Eq. (5) and Eq. (6) and their relation to mutual information, committed to adding full derivations in the appendix, and reported results with multiple random seeds.
* [ojQq] We elaborated on experimental design choices, justified our reconstruction-loss metric as a proxy for mutual information, added error bars through repeated multiple seeds, and clarified limitations and scope of applicability to more complex domains.
* Furthermore, we fixed typos, improved clarity, and committed to further editing and figure improvements.

Following these updates, we believe we have addressed most concerns and strengthened the paper’s contribution both as a theoretical framework for understanding LAMs, and insights into current practice. We hope the reviewers feel similarly.

---

### Decision · Program_Chairs · 2025-09-17

**Decision:**

Accept (poster)

**Comment:**

The paper presents a theoretical analysis of Latent Action Models (LAMs) using a linear framework. The analysis shows that LAMs learn representations of high-variance changes between video frames. This provides a basis for understanding why techniques such as data augmentation and auxiliary action prediction are effective.

The work is one of the first theoretical studies of LAMs. The primary concern from all reviewers was whether findings from the linear model would apply to the non-linear LAMs used in practice. During the rebuttal, the authors provided new experiments using a non-linear, CNN-based LAM. The results were consistent with the conclusions from the linear analysis.

The AC agrees with the consensus from reviewers, and recommends acceptance.